# On the Applicability of Quantum Machine Learning

**DOI:** 10.3390/e25070992

**Published:** 2023-06-28

**Authors:** Sebastian Raubitzek, Kevin Mallinger

**Affiliations:** 1Data Science Research Unit, TU Wien, Favoritenstrasse 9-11/194, 1040 Vienna, Austria; 2SBA Research gGmbH, Floragasse 7/5.OG, 1040 Vienna, Austria

**Keywords:** quantum machine learning, variational quantum circuit, quantum kernel estimator, Qiskit, Ridge, Lasso, XGBoost, LightGBM, CatBoost, classification, quantum computing, boost classifiers, neural networks

## Abstract

In this article, we investigate the applicability of quantum machine learning for classification tasks using two quantum classifiers from the Qiskit Python environment: the variational quantum circuit and the quantum kernel estimator (QKE). We provide a first evaluation on the performance of these classifiers when using a hyperparameter search on six widely known and publicly available benchmark datasets and analyze how their performance varies with the number of samples on two artificially generated test classification datasets. As quantum machine learning is based on unitary transformations, this paper explores data structures and application fields that could be particularly suitable for quantum advantages. Hereby, this paper introduces a novel dataset based on concepts from quantum mechanics using the exponential map of a Lie algebra. This dataset will be made publicly available and contributes a novel contribution to the empirical evaluation of quantum supremacy. We further compared the performance of VQC and QKE on six widely applicable datasets to contextualize our results. Our results demonstrate that the VQC and QKE perform better than basic machine learning algorithms, such as advanced linear regression models (Ridge and Lasso). They do not match the accuracy and runtime performance of sophisticated modern boosting classifiers such as XGBoost, LightGBM, or CatBoost. Therefore, we conclude that while quantum machine learning algorithms have the potential to surpass classical machine learning methods in the future, especially when physical quantum infrastructure becomes widely available, they currently lag behind classical approaches. Our investigations also show that classical machine learning approaches have superior performance classifying datasets based on group structures, compared to quantum approaches that particularly use unitary processes. Furthermore, our findings highlight the significant impact of different quantum simulators, feature maps, and quantum circuits on the performance of the employed quantum estimators. This observation emphasizes the need for researchers to provide detailed explanations of their hyperparameter choices for quantum machine learning algorithms, as this aspect is currently overlooked in many studies within the field. To facilitate further research in this area and ensure the transparency of our study, we have made the complete code available in a linked GitHub repository.

## 1. Introduction

Quantum computing has recently gained significant attention due to its potential to solve complex computational problems exponentially faster than classical computers [1]. Quantum machine learning (QML) is an emerging field that combines the power of quantum computing with traditional machine learning techniques to solve real-world problems more efficiently [2,3]. Various QML algorithms have been proposed, such as quantum kernel estimator [4] and variational quantum circuit [5,6], which have shown promising results in diverse applications, including pattern recognition and classification tasks [7,8,9].

In this study, we aim to compare QKE (quantum kernel estimator) and VQC (variational quantum circuit) with powerful classical machine learning methods such as XGBoost [10], Ridge [11], Lasso [12], LightGBM [13], CatBoost [14], and MLP (multilayer perceptron) [15] on six benchmark datasets partially available in the Scikit-learn library [16] as well as artificially generated datasets. To ensure a fair comparison on the benchmark datasets, we perform a randomized search to optimize hyperparameters for each algorithm, thereby providing a comprehensive statistical comparison of their performance. Furthermore, we provide the full program code in a GitHub repository [17] to make our results reproducible and boost research that can potentially build on our approach.

Since quantum machines are not readily accessible, we can only compare these algorithms’ performance on simulated quantum circuits. Although this approach does not reveal the full potential of quantum machine learning, it does highlight how the discussed quantum machine learning methods handle different levels of complexity inherent in the datasets. For this reason, we also developed a method to generate artificial datasets based on quantum mechanical concepts to provide a prototype for a particularly well-suited dataset for quantum machine learning. This will estimate the possible improvements that quantum machine learning algorithms can offer over classical methods in terms of accuracy and efficiency, considering the computational resources needed to simulate quantum circuits.

In this study, we address and partially answer the following research questions:How do QKE and VQC algorithms compare to classical machine learning methods such as XGBoost, Ridge, Lasso, LightGBM, CatBoost, and MLP regarding accuracy and efficiency on simulated quantum circuits?To what extent can a randomized search to find a suitable set of hyperparameters make the performance of quantum machine learning algorithms comparable to classical approaches?What are the limitations and challenges associated with the current state of quantum machine learning, and how can future research address these challenges to unlock the full potential of quantum computing in machine learning applications?Do quantum machine learning algorithms outperform regular machine learning algorithms on datasets constrained by the rules of quantum mechanics? Thus, do they provide a quantum advantage for datasets that exhibit strong symmetry properties in terms of adhering to Lie algebras?

The research presented in this article is partially inspired by the work of Zeguendry et al. [18], which offers an excellent review and introduction to quantum machine learning. However, their article does not delve into the tuning of hyperparameters for the quantum machine learning models employed, nor does it provide ideas on creating best-suited data for quantum machine learning classification tasks. We aim to expand the toolbox of quantum machine learning, first by discussing the space of Hyperparameters and second by providing a prototype for generating “quantum data”. Furthermore, this analysis will help determine the current state of quantum machine learning performance and whether researchers should employ these algorithms in their studies.

We provide the entire program code of our experiments and all the results in a GitHub repository, ensuring the integrity of our findings, fostering research in this field, and offering a comprehensive code for researchers to test quantum machine learning on their classification problems. Thereby, a key contribution of our research is not only the provision of a single implementation of a quantum machine learning algorithm, but also the execution of a randomized search for potential hyperparameters of both classical and quantum machine learning models and a novel approach for generating artificial classification problems based on concepts inherent to quantum mechanics, i.e., Lie groups and algebras.

This article is structured as follows: Section 2 discusses relevant and related work. In Section 3, we describe, reference, and, to some degree, derive all employed techniques. We will not discuss the mathematical details of all employed algorithms here, but rather refer the interested reader to the referenced sources. Section 4 describes our performed experiments in detail, followed by the obtained results in Section 5, which also features a discussion of our findings. Finally, we conclude our findings in Section 6.

## 2. Related Work

Considerable research was conducted in recent years to advance quantum machine learning environments and their application field. This starts in the data encoding process, in which Schuld and Killoran [3] investigated quantum machine learning in feature Hilbert spaces theoretically. They proposed a framework for constructing quantum embeddings of classical data to enable quantum algorithms that learn and classify data in quantum feature spaces.

Further research was conducted on introducing novel architectural frameworks. For this, Mitarai et al. [19] presented a method called quantum circuit learning (QCL), which uses parameterized quantum circuits to approximate classical functions. QCL can be applied to supervised and unsupervised learning tasks, as well as reinforcement learning.

Havlíček et al. [4] introduced a quantum-enhanced feature space approach using quantum circuits. This work demonstrated that quantum computers can effectively process classical data with quantum kernel methods, offering the potential for exponential speedup in certain applications.

Furthermore, Farhi and Neven [20] explored the use of quantum neural networks for classification tasks on near-term quantum processors. They showed that quantum neural networks can achieve good classification performance with shallow circuits, making them suitable for noisy intermediate-scale quantum (NISQ) devices.

Other research focused on the advancement of applying quantum fundamentals on classical machine learning applications. Hereby, Rebentrost et al. [21] introduced the concept of a quantum support vector machine for big data classification. They showed that the quantum version of the algorithm can offer exponential speedup compared to its classical counterpart, specifically in the kernel evaluation stage.

To advance the application field of quantum machine learning, Liu and Rebentrost [22] proposed a quantum machine learning approach for quantum anomaly detection. They demonstrated that their method can efficiently solve classification problems, even when the data have a high degree of entanglement.

In this regard, it is worth mentioning the work of Broughton et al. [23] introduced TensorFlow Quantum, an open-source library for the rapid prototyping of hybrid quantum-classical models for classical or quantum data. They demonstrated various applications of TensorFlow Quantum, including supervised learning for quantum classification, quantum control, simulating noisy quantum circuits, and quantum approximate optimization. Moreover, they showcased how TensorFlow Quantum can be applied to advanced quantum learning tasks such as meta-learning, layer-wise learning, Hamiltonian learning, sampling thermal states, variational quantum eigensolvers, classification of quantum phase transitions, generative adversarial networks, and reinforcement learning.

In the review paper by Zeguendry et al. [18], the authors present a comprehensive overview of quantum machine learning from the perspective of conventional machine learning techniques. The paper starts by exploring the background of quantum computing, its architecture, and an introduction to quantum algorithms. It then delves into several fundamental algorithms for QML, which form the basis of more complex QML algorithms and can potentially offer performance improvements over classical machine learning algorithms. In the study, the authors implement three machine learning algorithms: quanvolutional neural networks, quantum support vector machines, and variational quantum circuit. They compare the performance of these quantum algorithms with their classical counterparts on various datasets. Specifically, they implement quanvolutional neural networks on a quantum computer to recognize handwritten digits and compare its performance to convolutional neural networks, stating the performance improvements by quantum machine learning.

Despite these advancements, it is important to note that some of the discussed papers may not have used randomized search CV from Scikit-learn to optimize the classical machine learning algorithms, thereby overstating the significance of quantum supremacy. Nevertheless, the above-mentioned works present a comprehensive overview of the state of the art in quantum machine learning for classification, highlighting the potential benefits of using quantum algorithms in various forms and applications.

## 3. Methodology

This section presents our methodology for comparing the performance of classical and quantum machine learning techniques for classification tasks. Our approach is designed to provide a blueprint for future experiments in this area of research. We employ the Scikit-learn library, focusing on the inbuilt functions to select a good set of hyperparameters, i.e., RandomizedSearchCV to compare classical and quantum machine learning models. We also utilize the Qiskit library to incorporate quantum machine learning techniques into our experiments, [24]. The selected datasets for our study include both real-world and synthetic data, enabling a comprehensive evaluation of the classifiers’ performance.

### 3.1. Supervised Machine Learning

Supervised machine learning is a subfield of artificial intelligence that focuses on developing algorithms and models to learn patterns and make decisions or predictions based on data [25,26]. The main goal of supervised learning is to predict labels or outputs of new, unseen data given a set of known input–output pairs (training data). This section briefly introduces several classical machine learning techniques used for classification tasks, specifically in the context of supervised learning. These techniques serve as a baseline to evaluate the applicability of quantum machine learning approaches, which are the focus of this paper. Furthermore, we will then introduce the employed quantum machine learning algorithms.

One of the essential aspects of supervised machine learning is the ability to predict/classify data. The models are trained using a labeled dataset, and then the performance of the models is evaluated based on their accuracy in predicting the labels of previously unseen test samples [27]. This evaluation is crucial to estimate the model’s ability to generalize the learned information when making predictions on new, real-world data.

Various techniques, such as cross-validation and train-test splits, are often used to obtain reliable performance estimates of the models [28]. By comparing the performance of different models, researchers and practitioners can determine which model or algorithm is better suited for a specific problem domain.

### 3.2. Classical Supervised Machine Learning Techniques

The following list describes the employed algorithms that serve as a baseline for the afterwards described and later tested quantum machine learning algorithms.

**Lasso and Ridge Regression/Classification:** Lasso (least absolute shrinkage and selection operator) and Ridge Regression are linear regression techniques that incorporate regularization to prevent overfitting and improve model generalization [11,12]. Lasso uses L1 regularization, which tends to produce sparse solutions, while Ridge Regression uses L2 regularization, which prevents coefficients from becoming too large.Both of these regression algorithms can also be used for classification tasks.**Multilayer Perceptron:** MLP is a type of feedforward artificial neural network with multiple layers of neurons, including input, hidden, and output layers [15]. MLPs are capable of modeling complex non-linear relationships and can be trained using backpropagation.**Support Vector Machines (SVM):** SVMs are supervised learning models used for classification and regression tasks [29]. They work by finding the optimal hyperplane that separates the data into different classes, maximizing the margin between the classes.**Gradient Boosting Machines:** Gradient boosting machines are an ensemble learning method that builds a series of weak learners, typically decision trees, to form a strong learner [30]. The weak learners are combined by iteratively adding them to the model while minimizing a loss function. Notable gradient boosting machines for classification tasks include XGBoost [10], CatBoost [14], and LightGBM [13]. These three algorithms have introduced various improvements and optimizations to the original gradient boosting framework, such as efficient tree learning algorithms, handling categorical features, and reducing memory usage.

### 3.3. Quantum Machine Learning

Quantum machine learning is an emerging interdisciplinary field that leverages the principles of quantum mechanics and quantum computing to improve or develop novel algorithms for machine learning tasks [2]. This section introduces two key quantum machine learning techniques, Variational Quantum Circuit and Quantum Kernel Estimator, and discusses their connections to classical machine learning techniques. Additionally, we briefly introduce Qiskit Machine Learning, a Python package developed by IBM for implementing quantum machine learning algorithms. Furthermore, we want to mention the work done by [18] for a review of quantum machine learning algorithms and a more detailed discussion of the employed algorithms.

#### 3.3.1. Variational Quantum Circuit (VQC)

VQC is a hybrid quantum-classical algorithm that can be viewed as a quantum analog of classical neural networks, specifically the multilayer perceptron [5,6]. VQC employs a parametrized quantum circuit, which is trained using classical optimization techniques to find the optimal parameters for classification tasks. The learned quantum circuit can then be used to classify new data points.

Figure 1 illustrates the schematic depiction of the variational quantum circuit, which involves preprocessing the data, encoding it onto qubits using a feature map, processing it through a variational quantum circuit (Ansatz), measuring the final qubit states, and optimizing the circuit parameters θ, Thus, the main building blocks of the VQC are as follows:Preprocessing: The data are prepared and preprocessed before being encoded onto qubits.Feature map encoding (yellow in the figure): The preprocessed data are encoded onto qubits using a feature map.Variational quantum circuit (Ansatz) (steel-blue in the figure): The encoded data undergo processing through the variational quantum circuit, also known as the Ansatz, which consists of a series of quantum gates and operations.Measurement (orange in the figure): The final state of the qubits is measured, providing probabilities for the different quantum states.Parameter optimization (Optimizer): The variational quantum circuit is optimized by adjusting the parameters θ, such as the rotations of specific quantum gates, to improve the outcome/classification.

#### 3.3.2. Quantum Kernel Estimator

QKE is a technique that leverages the quantum computation of kernel functions to enhance the performance of classical kernel methods, such as support vector machines [4,31]. By computing the kernel matrix using quantum circuits, QKE can capture complex data relationships that may be challenging for classical kernel methods to exploit.

The main building blocks for the employed QKE, which are depicted in Figure 2 are as follows:Data preprocessing: The input data are preprocessed, which may include tasks such as data cleaning, feature scaling, or feature extraction. This step ensures that the data are in an appropriate format for the following quantum feature maps.Feature map encoding (yellow in the figure): The preprocessed data are encoded onto qubits using a feature map.Kernel computation (steel-blue in the figure): Instead of directly computing the kernel matrix from the original data, a kernel function is precomputed using the quantum computing capabilities, meaning that the inner product of two quantum states is estimated on a quantum simulator/circuit. This kernel function captures the similarity between pairs of data points in a high-dimensional feature space.SVM training: The precomputed kernel function is then used as input to the SVM algorithm for model training. The SVM aims to find an optimal hyperplane that separates the data points into different classes with the maximum margin.

Here, we need to mention that in the documentation of Qiskit machine learning, the developers provided a full QKE implementation without the need to use, e.g., Scikit-learn’s SVM-implementation. However, as of the writing of this article, this estimator is no longer available in Qiskit machine learning. Thus, one needs to use a support vector machine implementation from other sources after precomputing the kernel on a quantum simulator.

**Figure 2 entropy-25-00992-f002:**
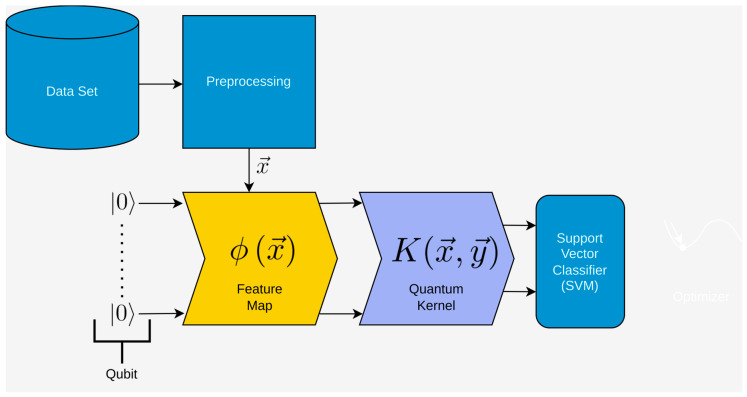
Schematic depiction of the quantum kernel estimator. The QKE consists of several steps. We colored the steps that are similar to classical support vector machines in light blue and the other steps in yellow and steel-blue. The employed QKE algorithm consists of a support vector machine algorithm with precomputed kernel, i.e., a classical machine learning method that leverages the power of quantum computing to efficiently compute the kernel matrix.

#### 3.3.3. Qiskit Machine Learning

Qiskit Machine Learning is an open-source Python package developed by IBM for implementing quantum machine learning algorithms [24]. This package enables researchers and practitioners to develop and test quantum machine learning algorithms, including VQC and QKE, using IBM’s quantum computing platform. It provides tools for building and simulating quantum circuits, as well as interfaces to classical optimization and machine learning libraries. Thus, we used this environment and the corresponding quantum simulators described in Appendix A for our experiments.

### 3.4. Accuracy Score for Classification

The accuracy score is a standard metric used to evaluate the performance of classification algorithms. We employed the accuracy score to evaluate all presented experiments. It is defined as the ratio of correct predictions to the total number of predictions. The formula for the accuracy score is defined as follows:(1)$$\mathrm{Accuracy}=\frac{\mathrm{Number}\;\mathrm{of}\;\mathrm{correct}\;\mathrm{predictions}}{\mathrm{Total}\;\mathrm{number}\;\mathrm{of}\;\mathrm{predictions}}$$

In Scikit-learn, the accuracy score can be computed using the accuracy_score function from the ‘sklearn.metrics’ module [16]. For more information on the accuracy score and its interpretation, refer to the Scikit-learn documentation [16].

### 3.5. Datasets

In this study, we used six classification datasets from various sources. Two datasets are part of the Scikit-learn library, while the remaining four are obtained/fetched from OpenML. The datasets are described below:**Iris Dataset:** A widely known dataset consisting of 150 samples of iris flowers, each with four features (sepal length, sepal width, petal length, and petal width) and one of three species labels (Iris Setosa, Iris Versicolor, or Iris Virginica). This dataset is included in the Scikit-learn library [16].**Wine Dataset:** A popular dataset for wine classification, which consists of 178 samples of wine, each with 13 features (such as alcohol content, color intensity, and hue) and one of three class labels (class 1, class 2, or class 3). This dataset is also available in the Scikit-learn library [16].**Indian Liver Patient Dataset (LPD):** This dataset contains 583 records, with 416 liver patient records and 167 non-liver patient records [32]. The dataset includes ten variables: age, gender, total bilirubin, direct bilirubin, total proteins, albumin, A/G ratio, SGPT, SGOT, and Alkphos. The primary task is to classify patients into liver or non-liver patient groups.**Breast Cancer Coimbra Dataset:** This dataset consists of 10 quantitative predictors and a binary dependent variable, indicating the presence or absence of breast cancer [33,34]. The predictors are anthropometric data and parameters obtainable from routine blood analysis. Accurate prediction models based on these predictors can potentially serve as a biomarker for breast cancer.**Teaching Assistant Evaluation Dataset:** This dataset includes 151 instances of teaching-assistant (TA) assignments from the Statistics Department at the University of Wisconsin-Madison, with evaluations of their teaching performance over three regular semesters and two summer semesters [35,36]. The class variable is divided into three roughly equal-sized categories (“low”, “medium”, and “high”). There are six attributes, including whether the TA is a native English speaker, the course instructor, the course, the semester type (summer or regular), and the class size.**Impedance Spectrum of Breast Tissue Dataset:** This dataset contains impedance measurements of freshly excised breast tissue at the following frequencies: 15.625, 31.25, 62.5, 125, 250, 500, and 1000 KHz [37,38]. The primary task is to predict the classification of either the original six classes or four classes by merging the fibro-adenoma, mastopathy, and glandular classes whose discrimination is not crucial.

These datasets were selected for their diverse domains and varied classification tasks, providing a robust testing ground for the quantum classifiers we employed in our experiments. Furthermore, we used artificially generated datasets to control the number of samples. Here, Scikit-learn provides a valuable function called make_classification to generate synthetic classification datasets. This function creates a random *n*-class classification problem, initially creating clusters of points normally distributed about vertices of an *n*-informative-dimensional hypercube, and assigns an equal number of clusters to each class [16]. It introduces interdependence between features and adds further noise to the data. The generated data are highly customizable, with options for specifying the number of samples, features, informative features, redundant features, repeated features, classes, clusters per class, and more. For more details on the make_classification function and its parameters, refer to the Scikit-learn documentation available on scikit-learn.org (accessed on 25 June 2023).

#### 3.5.1. Data Obtained from Lie-Algebras

We construct another artificial dataset final dataset for our final evaluation; however, this time, we do this by using tools from the theory of Lie groups. The reason for employing these concepts is that we want to produce data that resembles the complexity inherent to the Qubit-Vectorspace of quantum machine learning and that, furthermore, is generated by applying transformations on vectors that are similar to the manipulations present in quantum machine learning algorithms, e.g., for the VQC, rotations of/around the Bloch-sphere. Thus, overall, we aim to provide random data for a classification task to show a case where the authors assume quantum machine learning algorithms can, because of their inherent structure, outperform classical machine learning algorithms, and thus, provide a prototype on the type of data specifically tailored to address the inherent structure of quantum machine learning. The theoretical foundations of this section are obtained from [39], and thus, the interested reader is referred to this book for a profound introduction to Lie groups. To further explain the employed ideas, we start by introducing the concept of a Lie group *G* and the corresponding Lie-algebra g.

A Lie group is a mathematical structure that captures the essence of continuous symmetry. Named after the Norwegian mathematician Sophus Lie, Lie groups are ubiquitous in many areas of mathematics and physics, including the study of differential equations, geometry, and quantum mechanics.

A Lie group is a set *G* that has the structure of both a smooth manifold and a group in such a way that the group operations (multiplication and inversion) are smooth. That is, a Lie group is a group that is also a differentiable manifold, such that the group operations are compatible with the smooth structure.

Thus, a Lie group is a set *G* equipped with a group structure (i.e., a binary operation G×G→G, (g,h)↦gh that is associative, an identity element e∈G, and an inversion operation G→G, $g\mapsto g^{-1}$) and a smooth manifold structure such that the following conditions are satisfied:The multiplication map μ:G×G→G defined by μ(g,h)=gh is smooth.The inversion map ι:G→G defined by $\iota \left(g\right)=g^{-1}$ is smooth.

Lie algebra is associated with each Lie group, a vector space equipped with a binary operation called the Lie bracket. The Lie algebra captures the local structure of the Lie group near the identity element, meaning that the Lie algebra of a Lie group *G* is the tangent space at the identity, denoted $T_{e}G$, equipped with the Lie bracket operation. The Lie bracket is defined in terms of the group operation and the differential.

There is a map from the Lie algebra to the Lie group called the exponential map, denoted $\exp:T_{e}G\to G$. The exponential map provides a way to generate new group elements from elements of the Lie algebra. In particular, given an element *X* of the Lie algebra, exp(X) is a group element close to the identity if *X* is ‘small’. We will exploit this concept to generate random data associated with a specific group:

We start with a set of generators $T_{a}$ contained within the Lie-algebra *g* of a Lie group *G*, where $a=1,2,\cdots d_{g}$, i.e., the dimension of the Lie-algebra. We can then create elements g∈G by employing:(2)$$g=e^{i\sum_{a}\theta_{a}T_{a}}\;,\;\mathrm{where}\theta_{a}\in \left[0,2\pi \right]\;.$$

We used the condition for our $\theta_{a}$-values without loss of generality due to the periodicity of the exponential function. To generate our random data, we randomly choose our $\theta_{a}$ and create an element of our group. We then apply this element to a corresponding base vector of our vector space.

Specifically, in our example, we use the Lie-group SU(2). The special unitary group of degree 2, denoted as SU(2), is a Lie group of 2 × 2 unitary matrices with determinant 1.
(3)$$\mathrm{SU}\left(2\right)=\left\{U\in C^{2\times 2}:UU^{†}=I,\;\det\left(U\right)=1\right\}$$

The corresponding Lie algebra, su(2), consists of 2 × 2 Hermitian traceless matrices, i.e., the Pauli matrices:(4)$$\sigma_{1}=\left(\begin{array}{cc} 0 & 1\\ 1 & 0 \end{array}\right),\;\sigma_{2}=\left(\begin{array}{cc} 0 & -i\\ i & 0 \end{array}\right),\;\sigma_{3}=\left(\begin{array}{cc} 1 & 0\\ 0 & -1 \end{array}\right)$$

The commutation relations of the Pauli matrices form the structure of the su(2) Lie algebra:(5)$$\left[\sigma_{i},\sigma_{j}\right]=2i\varepsilon_{ijk}\sigma_{k}$$
where [·,·] denotes the commutator and $\varepsilon_{ijk}$ is the Levi-Civita symbol.

To generate a classification dataset from this algebra, we use the following procedure:Find a set of random parameters $\theta \in \left(0;\pi \right],\;\phi \in \left(0;2\pi \right],\;\lambda \in \left(0;2\pi \right]$;2.We then create an element *U* of SU(2) using these these randomly set parameters: $U=e^{i\left(\theta \sigma_{1}+\phi \sigma_{2}+\lambda \sigma_{3}\right)}$;3.Next, we take one of the basevectors from $C^{2}$, denoted as $\hat{v}$ to create a new complex vector $\vec{v}$ using the previously obtained matrix *U* such that: $\vec{v}=U\cdot \hat{v}$;4.This vector is then separated into four features $F_{j}$ such that:
(6)F1=Rev1
(7)F2=Imv1
(8)F3=Rev2
(9)F4=Imv2,
where $v_{1}$ and $v_{2}$ denotes the individual components of the vector $\vec{v}$, and Re[…] and Im[…] denote their respective real and imaginary parts;5.Finally, we assign a class label *C* to this collection of features such that:
(10)$$C=\left\{\begin{array}{cc} 0 & \mathrm{if}\;\theta <\frac{\pi }{2}\\ 1 & \mathrm{if}\;\theta >\frac{\pi }{2} \end{array}\right.\;,$$
and collect the features and the class label into one sample $\left[F_{1},F_{2},F_{3},F_{4},C\right]$. We repeat this process $N_{S}$ times, starting with 1, where $N_{S}$ is the number of samples that we want for our dataset.

Note that this approach can be extended to arbitrary Lie groups, given that one can construct or obtain a Lie group’s generators.

## 4. Experimental Design

In this section, we describe our experimental design, which aims to provide a fair and comprehensive comparison of the performance of classical machine learning (ML) and quantum machine learning techniques, as discussed in Section 3.2 and Section 3.3. Our experiments involve two main components: Firstly, assessing the algorithms’ performance on artificially generated datasets with varying parametrizations, and secondly, evaluating the algorithms’ performance on benchmark datasets using randomized search to optimize hyperparameters, ensuring a fair comparison. By carefully selecting our experimental setup, we avoid the issue of “cherry-picking” only a favorable subset of results, a common problem in machine learning, leading to heavily biased conclusions.

### 4.1. Artificially Generated Scikit Datasets

To generate the synthetic classification dataset, we utilized Scikit-learn’s make_classification function. We employed two features and two classes while varying the number of samples to obtain a performance curve illustrating how the chosen algorithms’ performance changes depending on the sample size.

We partitioned each dataset such that 20% of the original data were reserved as a test set to evaluate the trained algorithm, producing the accuracy score used for our assessment. Furthermore, each dataset was normalized such that all features are within the unit interval $\left[0,1\right]$.

As a baseline, we employed the seven classical machine learning algorithms described in Section 3.2, namely Lasso, Ridge, MLP, SVM, XGBoost, LightGBM, and CatBoost. We used two different parameterizations for the classical machine learning algorithms for our comparisons. Firstly, we applied the out-of-the-box implementation without any hyperparameter optimization. Secondly, we used an optimized version of each algorithm found through Scikit-learn’s RandomizedSearchCV by testing 20 different models.

We then examined 20 distinct parameter configurations, each for the VQC and QKE classifiers, randomly selected from a predefined parameter distribution. Appendix A discusses the parameter grids for all utilized algorithms and all experiments.

### 4.2. Artificially Generated SU(2) Datasets

For our synthetic SU(2) classification dataset, we used the concepts previously discussed in Section 3.5.1. We employed two complex features, i.e., resulting in four continuous real features, and two classes while varying the number of samples to obtain a performance curve illustrating how the chosen algorithms’ performance changes depending on the sample size.

We partitioned each dataset such that 20% of the original data were reserved as a test set to evaluate the trained algorithm, producing the accuracy score used for our assessment. Furthermore, each dataset was normalized such that all features are within the unit interval $\left[0,1\right]$.

As a baseline, we employed the seven classical machine learning algorithms described in Section 3.2, namely Lasso, Ridge, MLP, SVM, XGBoost, LightGBM, and CatBoost. We used two different parameterizations for the classical machine learning algorithms for our comparisons. Firstly, we applied the out-of-the-box implementation without any hyperparameter optimization. Secondly, we used an optimized version of each algorithm found through Scikit-learn’s RandomizedSearchCV by testing 20 different models.

We then examined 20 distinct parameter configurations, each for the VQC and QKE classifiers, randomly selected from a predefined parameter distribution. Appendix A discusses the parameter grids for all utilized algorithms and all experiments.

### 4.3. Benchmark Datasets and Hyperparameter Optimization

Our last experiment was to test the two employed quantum machine learning algorithms against the classical machine learning algorithms on six benchmark datasets (Section 3.5). For this reason, we employed Scikit-learn’s RandomizedSearchCV to test 20 randomly parameterized models for each algorithm to report the best of these tests. Again, we used a train-test-split to keep 20% of the original data to test the trained algorithm. Furthermore, each dataset was normalized such that all features are within the unit interval $\left[0,1\right]$.

## 5. Results

In this section, we present the results of our experiments, comparing the performance of classical machine learning and quantum machine learning techniques on both artificially generated datasets and benchmark datasets (Section 3.5). By analyzing the results, we aim to draw meaningful insights into the strengths and weaknesses of each approach and provide a blueprint for future studies in the area. Everything was calculated on a Lenovo ThinkCentre machine using an *Intel(R) Core(TM) i7-4770 CPU 3.40GHz* and 16GB RAM and Linux 20.04. We used python 3.6 and the included packages are the following:


numpy version: 1.18.5

sklearn version: 0.23.1

catboost version: 0.26.1

xgboost version: 1.2.1

lightgbm version: 3.2.1
qiskit version: {‘qiskit-terra’: ‘0.19.2’, ‘qiskit-aer’: ‘0.10.3’, ‘qiskit-ignis’: ‘0.7.0’, ‘qiskit-ibmq-provider’: ‘0.18.3’, ‘qiskit-aqua’: None, ‘qiskit’: ‘0.34.2’, ‘qiskit-nature’: ‘0.3.1’, ‘qiskit-finance’: None, ‘qiskit-optimization’: None, ‘qiskit-machine-learning’: ‘0.3.1’}
qiskit_machine_learning version: 0.3.1


### 5.1. Performance on Artificially Generated Scikit Datasets

In this section, we compare the performance of quantum machine learning algorithms and classical machine learning algorithms on artificially generated classification datasets. The comprehensive experimental setup can be found in Section 4.1.

Regarding accuracy and runtime, our findings are presented in Table 1 and Table 2 and Figure 3, Figure 4 and Figure 5. The measured runtime includes hyperparameter tuning via randomized search and five-fold cross-validating, training, and testing the model.

While QML algorithms perform reasonably well, we observe that they are not a match for properly trained and/or sophisticated state-of-the-art classifiers. Even out-of-the-box implementations of state-of-the-art ML algorithms outperform QML algorithms on these artificially generated classification datasets.

The accuracy of the algorithms varies depending on the dataset size, with larger datasets posing more challenges. CatBoost performed best in our experiments, both out-of-the-box and when optimized in terms of high accuracy over all experiments. The quantum kernel estimator is the fifth-best algorithm overall in terms of accuracy, though it outperforms CatBoost regarding the runtime for CatBoost’s optimized version. XGBoost and support vector classification (SVC) follow closely, with competitive performances in terms of accuracy. However, variational quantum circuit struggles to achieve high accuracy compared to sophisticated boosting classifiers or support vector machines. Furthermore, we observe the best performance in terms of runtime for the two linear models, Lasso and Ridge. We need to point out that Lasso and Ridge both feature increased runtimes for the datasets of size 50; this is most likely due to the optimizer needing an increased number of iterations due to the small number of samples and their relatively scattered distribution of data points.

Other algorithms, such as multilayer perceptron, Ridge regression, Lasso regression, and LightGBM, exhibit varying performances depending on dataset size and optimization. Despite some reasonable results from QKE, we conclude that classical ML algorithms, particularly sophisticated boosting classifiers, should be chosen to tackle similar problems due to their ease of implementation, better runtime, and overall superior performance.

In summary, while QML algorithms have shown some promise, they cannot yet compete with state-of-the-art classical ML algorithms on artificially generated classification datasets in terms of accuracy and runtime.

### 5.2. Performance on Artificially Generated SU2 Datasets

In this section, we compare the performance of quantum machine learning algorithms and classical machine learning algorithms on artificially generated classification datasets based on Lie group structures. The detailed experimental setup can be found in Section 4.2.

Regarding accuracy and runtime, our findings are presented in Table 3 and Table 4 and Figure 6, Figure 7 and Figure 8. While QML algorithms perform reasonably well, we observe that they are not a match for properly trained and/or sophisticated state-of-the-art classifiers. Even out-of-the-box implementations of state-of-the-art ML algorithms outperform QML algorithms on artificially generated classification datasets that are particularly suited for QML.

The accuracy of the algorithms varies depending on the dataset size, with larger datasets providing increased accuracy for most algorithms. CatBoost performed best in our experiments, both out-of-the-box and when optimized in terms of high accuracy over all experiments. The quantum kernel estimator is the fifth-best algorithm overall in terms of accuracy. However, we observe that, on average, CatBoost with improved hyperparameters performs best over all experiments, but is outperformed by the best QKE implementation for 100 and 500 data points. Thus, we conclude that quantum kernel estimators can capture the complexity of this SU(2)-generated dataset, but overall, one is better off with an out-of-the-box CatBoost implementation. This means that we do not observe a quantum advantage for this type of data, but rather that the employed quantum kernel estimator behaves similarly to classical machine learning algorithms, i.e., it exhibits reasonable performance but does not perform best for all datasets, even the ones created by exploiting quantum symmetry properties.

Other algorithms, such as multilayer perceptron, Ridge regression, Lasso regression, and LightGBM, exhibit varying performances depending on dataset size and optimization. Despite some reasonable results from QKE, we conclude that classical ML algorithms, particularly sophisticated boosting classifiers, should be chosen to tackle similar problems due to their ease of implementation, better runtime, and overall superior performance. Furthermore, we again observe the best performance in terms of runtime for the two linear models, Lasso and Ridge. Moreover, again, we observe that Lasso and Ridge both feature increased runtimes for the datasets of size 50.

In summary, while QML algorithms have shown some promise, they cannot yet compete with state-of-the-art classical ML algorithms even on these SU(2)-datasets, where the authors intended to provide evidence for the quantum advantage for datasets generated from symmetry properties inherent to quantum mechanics.

### 5.3. Results on Benchmark Datasets

In this section, we discuss the performance of quantum machine learning and classical machine learning algorithms on six benchmark datasets described in Section 3.5. We include results for the quantum classifiers detailed in Section 3.3 and the classical machine learning classifiers discussed in Section 3.2. The scores/accuracies were obtained using randomized search cross-validation from Scikit-learn with 20 models and five-fold cross-validation.

Our results, shown in Table 5, display the best five-fold cross-validation scores (upper table) and the scores of the best model evaluated on an unseen test subset of the original data (lower table), which makes up 20% of the original data. We observe varying performances of the algorithms on these benchmark datasets.

Notably, both the variational quantum circuit and the quantum kernel estimator classifier show competitive performance on several datasets but do not consistently outperform classical ML algorithms. In particular, QKE achieves a perfect score on the Iris dataset, but its performance varies across the other datasets.

Classical ML algorithms, such as multilayer perceptron, support vector machines, XGBoost, LightGBM, and CatBoost, exhibit strong performance across all datasets, with some algorithms achieving perfect scores on multiple datasets. CatBoost consistently performs well, ranking as the top-performing algorithm on three of the six datasets. Ridge and Lasso regression show high accuracy on Iris and Wine datasets but perform poorly on the others.

When comparing the runtimes of the experiments, as presented in Table 6, it becomes evident that QML algorithms take substantially longer to execute than their classical counterparts. For instance, the VQC and QKE classifiers take hours to days to complete on various datasets, whereas classical ML algorithms such as Ridge, Lasso, MLP, SVM, XGBoost, LightGBM, and CatBoost typically take seconds to minutes.

This significant difference in runtimes could be attributed to the inherent complexity and resource requirements of QML algorithms, which generally demand specialized quantum hardware and simulators. On the other hand, classical ML algorithms are optimized for execution on conventional hardware, making them more efficient and faster to run.

In conclusion, while QML algorithms such as VQC and QKE demonstrate potential in achieving competitive performance on certain datasets, their relatively longer runtimes and less consistent performance across the benchmark datasets may limit their practical applicability compared to classical ML algorithms. Classical ML algorithms, such as CatBoost, XGBoost, and LightGBM, continue to offer superior and more consistent performance with faster execution times, solidifying their place as reliable and powerful tools for classification tasks.

### 5.4. Comparison and Discussion

In this study, we have compared the performance of quantum machine learning and classical machine learning algorithms on six benchmark datasets and two types of artificially generated classification datasets. We included results for quantum classifiers, such as variational quantum circuit and quantum kernel estimator, and classical machine learning classifiers, such as CatBoost, XGBoost, and LightGBM. Our experiments showed that while QML algorithms demonstrate potential in achieving competitive performance on certain datasets, they do not consistently outperform classical ML algorithms. Additionally, their longer runtimes for the whole process, i.e., hyperparameter tuning via randomized search and five-fold cross-validation, the corresponding training and testing, and less consistent performance across the benchmark datasets, may limit their practical applicability compared to classical ML algorithms, which continue to offer superior and more consistent performance with faster execution times. Furthermore, we constructed artificial datasets with the structure and rulings of quantum Mechanics in mind, i.e., we used symmetry properties and unitary transformations to generate a classification dataset from SU(2)-matrices in order to demonstrate an advantage of quantum machine learning algorithms to tackle problems with an inherent structure relatable to that of quantum circuits and quantum mechanics overall. However, also for these datasets, the employed quantum machine learning algorithms performed reasonably but did not outperform sophisticated boost classifiers. Thus, we cannot conclude a quantum advantage for these datasets.

It is essential to highlight that the QML algorithms’ performance in our experiments was based on simulated quantum infrastructures. This is a significant limitation to consider, as the specific constraints and characteristics of the simulated hardware may influence the performance of these algorithms. Furthermore, given the rapid advancement of quantum technologies and hardware, this constraint might be obsolete in the near future.

The impact of quantum simulators, feature maps, and quantum circuits on the performance of quantum estimators stems from the fact that these components play crucial roles in shaping the behavior and capabilities of quantum machine learning algorithms. Quantum simulators, which emulate quantum systems on classical computers, introduce various levels of approximation and noise, leading to deviations from ideal quantum behavior. Different simulators may employ distinct algorithms and techniques, resulting in variations in performance.

Feature maps, responsible for encoding classical data into quantum states, determine how effectively the quantum system can capture and process information. The choice of feature map can greatly influence the ability of quantum algorithms to extract meaningful features and represent the data in a quantum-mechanical space.

Similarly, quantum circuits, composed of quantum gates and operations, define the computational steps performed on the encoded data. Different circuit designs and configurations can affect the expressiveness and depth of the quantum computation, potentially impacting the accuracy and efficiency of the quantum estimators.

Considering the diverse options for quantum simulators, feature maps, and quantum circuits, it becomes essential for researchers to provide detailed explanations of their hyperparameter choices. This entails clarifying the rationale behind selecting a specific simulator, feature map, or circuit design, as well as the associated parameters and their values. By providing such explanations, researchers can enhance the reproducibility and comparability of results, enabling the scientific community to better understand the strengths and limitations of different quantum machine learning algorithms.

Unfortunately, the current state of the field often overlooks the thorough discussion of hyperparameter choices in many studies. This omission restricts the transparency and interpretability of research outcomes and hinders the advancement of quantum machine learning. To address this issue, researchers should embrace a culture of providing comprehensive documentation regarding hyperparameter selection, sharing insights into the decision-making process, and discussing the potential implications of different choices.

By encouraging researchers to provide detailed explanations of hyperparameter choices and corresponding code, we can foster a more robust and transparent research environment in quantum machine learning. This approach enables the replication and comparison of results, promotes knowledge sharing, and ultimately contributes to the development of reliable and effective quantum machine learning algorithms. Additionally, our program code serves as introductory material, providing easy-to-use implementations and a foundation for comparing quantum machine learning and classical machine learning (CML) algorithms.

One possible direction for future research is exploring quantum ensemble classifiers and, consequently, quantum boosting classifiers, as suggested by Schuld et al. [40]. This approach might help in improving the capabilities of QML algorithms and make them more competitive with state-of-the-art classical ML algorithms in terms of high accuracies.

Finally, the relatively lower performance of the employed quantum machine learning algorithms compared to, for example, the employed boosting classifiers might be attributed to quantum machine learning, being constrained by specific rules of quantum mechanics.

In the authors’ opinion, quantum machine learning might be constrained by the unitary transformations inherent in, for example, the variational quantum circuits. These transformations are part of the unitary group U(n). Thus, all transformations are constrained by symmetry properties. Classical machine learning models are not constrained by these limitations, meaning that, for instance, different activation functions in neural networks do not preserve certain distance metrics or probabilities when processing data. However, expanding the set of transformations of quantum machine learning and getting rid of possible constraints might improve the capabilities of quantum machine learning models such that these algorithms might be better capable of capturing the information of more complex data. However, this needs to be discussed in the context of quantum computers such that one determines what all possible transformations on a quantum computer are. This means that future research needs to consider the applicability of advanced mathematical frameworks for quantum machine learning regarding the formal requirements of quantum computers.

Furthermore, another constraint of quantum machine learning is that it, and quantum mechanics in general, relies on Hermitian matrices, e.g., to provide real-valued eigenvalues of observables. However, breaking this constraint might be another way to broaden the capabilities of quantum machine learning to better capture complexity, e.g., by using non-Hermitian kernels in a quantum kernel estimator. Here, we want to mention the book by Moiseyev [41], which introduces non-Hermitian quantum mechanics. Furthermore, quantum computers, in general, might provide a testing ground for non-Hermitian quantum mechanics in comparison to Hermitian quantum mechanics. However, at this point, this is rather speculative, but given that natural data are nearly always corrupted by noise and symmetries are never truly perfect in nature, breaking constraints and symmetries might be ideas to expand the capabilities of QML.

## 6. Conclusions

In this research, we have explored the applicability of quantum machine learning for classification tasks by examining the performance of variational quantum circuit and quantum kernel estimator algorithms. Our comparison of these quantum classifiers with classical machine learning algorithms, such as XGBoost, Ridge, Lasso, LightGBM, CatBoost, and MLP, on six benchmark datasets and artificially generated classification datasets demonstrated that QML algorithms can achieve competitive performance on certain datasets. However, they do not consistently outperform their classical ML counterparts, particularly with regard to runtime performance and accuracy. Quite the contrary, classical machine learning algorithms still demonstrate superior performance, especially in terms of increased accuracy, in most of our experiments. Furthermore, we cannot conclude a quantum advantage even for artificial data built by data manipulations inherent to quantum mechanics.

As our study’s performance comparison relied on simulated quantum circuits, it is important to consider the limitations and characteristics of simulated hardware, which may affect the true potential of quantum machine learning. Given the rapid advancement of quantum technologies and hardware, these constraints may become less relevant in the future.

Quantum simulators, feature maps, and quantum circuits significantly influence quantum estimator performance; hence, a detailed discussion of the chosen hyperparameters is essential. The absence of such a discussion in current research limits the interpretation and replication of experiments. Thus, we aim to encourage transparency in decision-making processes to promote a robust research environment, aiding in knowledge sharing and the creation of reliable quantum machine learning algorithms.

Despite the current limitations, this study has shed light on the potential and challenges of quantum machine learning compared to classical approaches. Thus, by providing our complete code in a GitHub repository, we hope to foster transparency, encourage further research in this field, and offer a foundation for other researchers to build upon as they explore the world of quantum machine learning. Furthermore, the developed SU(2)-data creation might serve as a quantum data prototype for future experiments, and both quantum and regular machine learning algorithms can be tested for their accuracy on datasets like these.

Future research should also consider exploring quantum ensemble classifiers and quantum boosting classifiers, as well as addressing the limitations imposed by the specific rules of quantum mechanics. By breaking constraints and symmetries and expanding the set of transformations in quantum machine learning, researchers may be able to unlock its full potential.

## Figures and Tables

**Figure 1 entropy-25-00992-f001:**
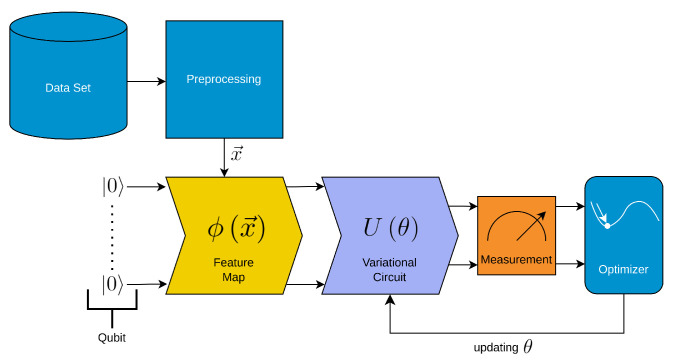
Schematic depiction of the variational quantum circuit. The VQC consists of several steps. We colored the steps that are similar to classical neural networks in light blue and the other steps in yellow, steel-blue, and orange.

**Figure 3 entropy-25-00992-f003:**
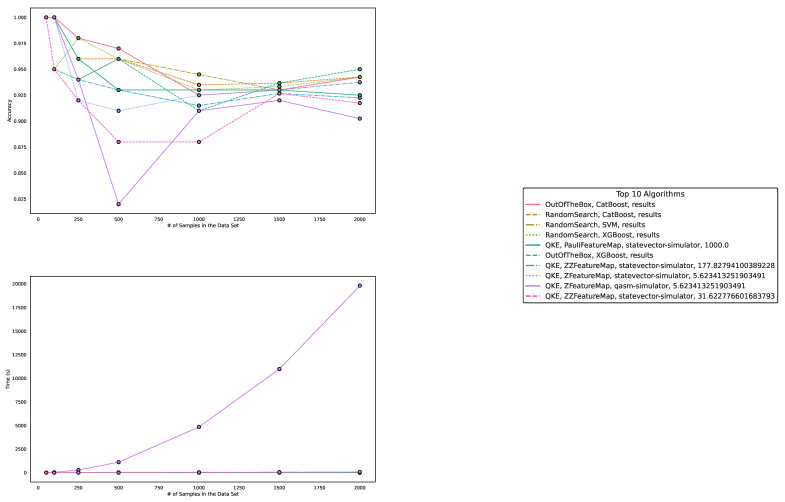
These figures depict the results from our experiments, comparing the five best QML and classical ML algorithms on artificially generated datasets in terms of accuracy. The **upper part** illustrates the accuracy of the algorithms on different sample sizes, while the **lower part** demonstrates how the runtimes change with increasing size of the test dataset. The **right part** contains the legend, indicating which algorithms were used, and more specifically, the different parametrizations of the employed quantum machine learning algorithms. Furthermore, the legend is sorted in decreasing order of the average accuracy of the employed algorithms. The parametrization for the QKE is as follows: QKE, feature map, quantum simulator, C-Value for the SVM algorithm.

**Figure 4 entropy-25-00992-f004:**
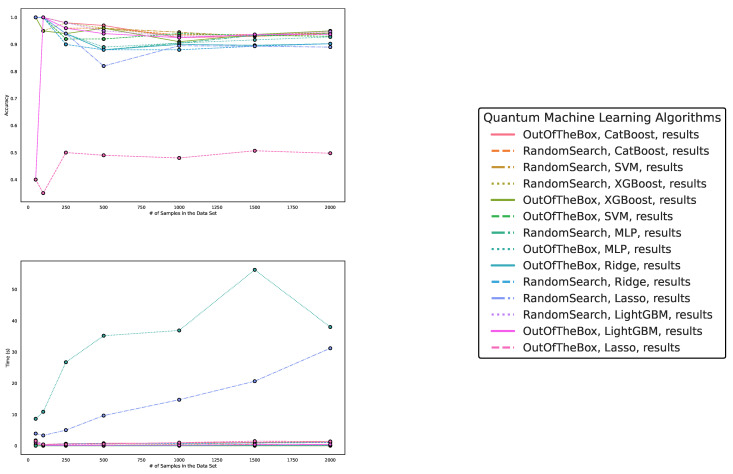
These figures depict the results from our experiments, comparing differently parameterized classical machine learning algorithms on artificially generated datasets. The **upper part** illustrates the behavior of the accuracies, while the **lower part** demonstrates how the run times change with the increasing size of the test dataset. The **right part** contains the legend, indicating which algorithms were used, and more specifically, the different parametrizations of the employed machine learning algorithms. Furthermore, the legend is sorted in decreasing order of the average accuracy of the employed algorithms.

**Figure 5 entropy-25-00992-f005:**
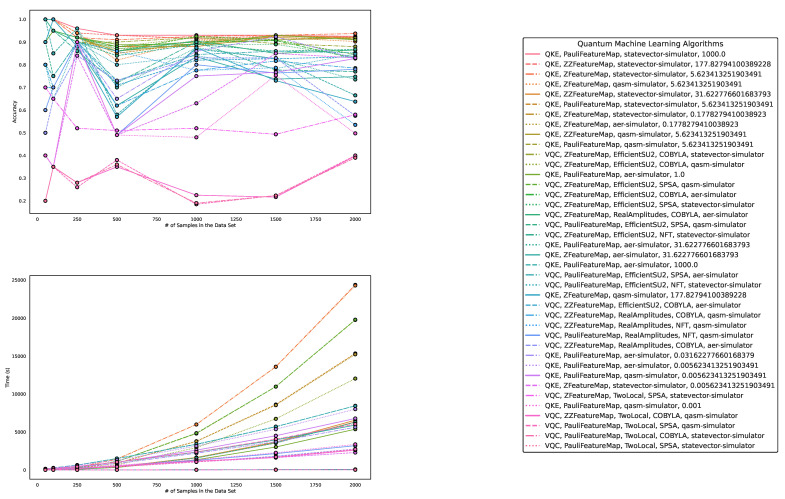
These figures depict the results from our experiments for the artificially generated datasets, comparing differently parameterized QML algorithms on artificially generated datasets. The **upper part** illustrates the behavior of the accuracies, while the **lower part** demonstrates how the runtimes change with the increasing size of the test datasets. The **right part** contains the legend, indicating which algorithms were used, and more specifically, the different parametrizations of the employed quantum machine learning algorithms. Furthermore, the legend is sorted in decreasing order of the average accuracy of the employed algorithms. The parametrization for the QKE is as follows: QKE, feature map, quantum simulator, C-Value for the SVM algorithm. The parametrization for the VQC is as follows: VQC, feature map, Ansatz, optimizer, quantum simulator.

**Figure 6 entropy-25-00992-f006:**
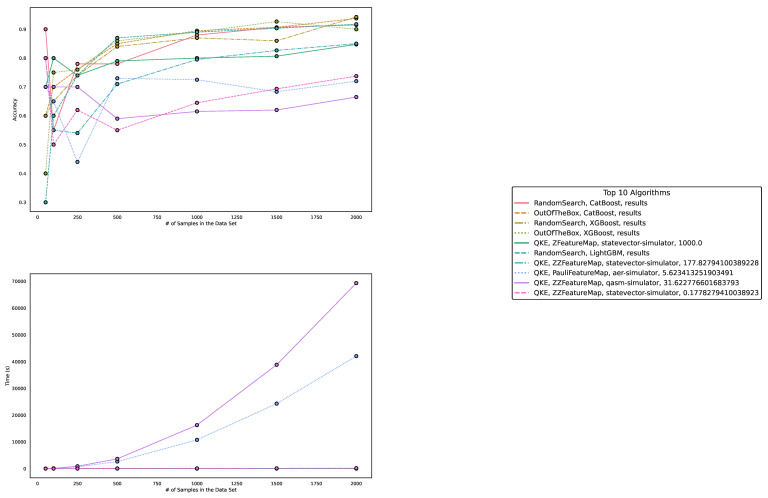
These figures depict the results from our experiments, comparing the five best QML and classical ML algorithms in terms of accuracy on datasets using the exponential map to create SU(2)-transformations on complex vectors. The **upper part** illustrates the accuracy of the algorithms on different sample sizes, while the **lower part** demonstrates how the runtimes change with the increasing size of the test dataset. The **right part** contains the legend, indicating which algorithms were used, and, more specifically, the different parametrizations of the employed quantum machine learning algorithms. Furthermore, the legend is sorted in decreasing order of the average accuracy of the employed algorithms. The parametrization for the QKE is as follows: QKE, feature map, quantum simulator, C-Value for the SVM algorithm.

**Figure 7 entropy-25-00992-f007:**
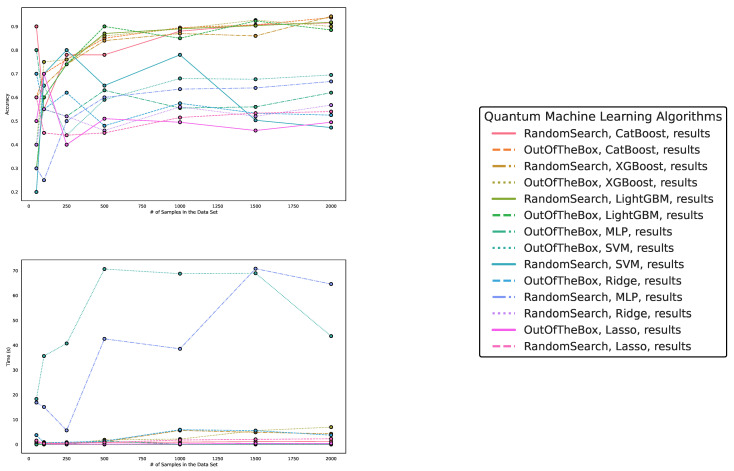
These figures depict the results from our experiments, comparing differently parameterized classical machine learning algorithms on the SU(2)-generated datasets. The **upper part** illustrates the behavior of the accuracies, while the **lower part** demonstrates how the run times change with the increasing size of the test dataset. The **right part** contains the legend, indicating which algorithms were used, and more specifically, the different parametrizations of the employed machine learning algorithms. Furthermore, the legend is sorted in decreasing order of the average accuracy of the employed algorithms.

**Figure 8 entropy-25-00992-f008:**
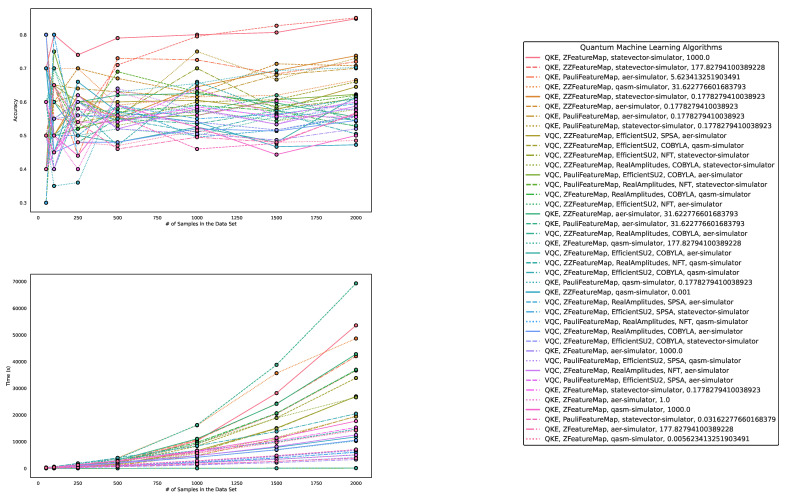
These figures depict the results from our experiments for the artificially generated datasets, comparing differently parameterized QML algorithms on the SU(2)-generated datasets. The **upper part** illustrates the behavior of the accuracies, while the **lower part** demonstrates how the runtimes change with the increasing size of the test datasets. The **right part** contains the legend, indicating which algorithms were used and, more specifically, the different parametrizations of the employed quantum machine learning algorithms. Furthermore, the legend is sorted in decreasing order of the average accuracy of the employed algorithms. The parametrization for the QKE is as follows: QKE, feature map, quantum simulator, C-Value for the SVM algorithm. The parametrization for the VQC is as follows: VQC, feature map, Ansatz, optimizer, quantum simulator.

**Table 1 entropy-25-00992-t001:** This table presents the scores/accuracies of our experiments conducted on artificially generated classification datasets of varying sizes, e.g., 50 and 100. Given these different dataset sizes, this table is sorted in decreasing order of the average accuracy over all different sample sizes of each algorithm. The parametrization for the QKE is as follows: QKE, feature map, quantum simulator, C-Value for the SVM algorithm. The parametrization for the VQC is as follows: VQC, feature map, Ansatz, optimizer, quantum simulator. For the classical machine learning algorithms, *OutOfTheBox* means that we did not tune the hyperparameters of the employed algorithm and *RandomSearch* refers to hyperparameter optimization via a randomized search.

Algorithm/Parametrization	Size 50	Size 100	Size 250	Size 500	Size 1000	Size 1500	Size 2000	Average
OutOfTheBox, CatBoost, results	1.0	1.0	0.98	0.97	0.925	0.93	0.9425	0.963929
RandomSearch, CatBoost, results	1.0	1.0	0.96	0.96	0.935	0.936667	0.9425	0.962024
RandomSearch, SVM, results	1.0	1.0	0.94	0.96	0.945	0.93	0.9375	0.958929
RandomSearch, XGBoost, results	1.0	0.95	0.98	0.96	0.93	0.933333	0.9425	0.956548
QKE, PauliFeatureMap, statevector-simulator, 1000.0	1.0	1.0	0.96	0.93	0.93	0.93	0.925	0.953571
OutOfTheBox, XGBoost, results	1.0	0.95	0.94	0.96	0.91	0.936667	0.95	0.949524
OutOfTheBox, SVM, results	1.0	1.0	0.92	0.92	0.94	0.933333	0.93	0.949048
QKE, ZZFeatureMap, statevector-simulator, 177.82794100389228	1.0	1.0	0.94	0.93	0.915	0.926667	0.9225	0.947738
QKE, ZFeatureMap, statevector-simulator, 5.623413251903491	1.0	1.0	0.92	0.91	0.925	0.93	0.9375	0.946071
RandomSearch, MLP, results	1.0	1.0	0.94	0.88	0.905	0.933333	0.94	0.942619
OutOfTheBox, MLP, results	1.0	1.0	0.94	0.89	0.905	0.916667	0.9275	0.939881
OutOfTheBox, Ridge, results	1.0	1.0	0.94	0.88	0.9	0.896667	0.9025	0.93131
QKE, ZFeatureMap, qasm-simulator, 5.623413251903491	1.0	1.0	0.94	0.82	0.91	0.92	0.9025	0.9275
QKE, ZZFeatureMap, statevector-simulator, 31.622776601683793	1.0	0.95	0.92	0.88	0.88	0.926667	0.9175	0.924881
QKE, PauliFeatureMap, statevector-simulator, 5.623413251903491	1.0	0.95	0.92	0.85	0.895	0.93	0.92	0.923571
QKE, ZFeatureMap, statevector-simulator, 0.1778279410038923	1.0	0.95	0.9	0.88	0.9	0.92	0.9125	0.923214
QKE, ZFeatureMap, aer-simulator, 0.1778279410038923	1.0	0.95	0.9	0.87	0.905	0.92	0.9125	0.9225
RandomSearch, Ridge, results	1.0	1.0	0.9	0.88	0.88	0.893333	0.9025	0.922262
QKE, ZZFeatureMap, qasm-simulator, 5.623413251903491	1.0	0.95	0.92	0.86	0.89	0.91	0.9175	0.921071
QKE, PauliFeatureMap, qasm-simulator, 5.623413251903491	1.0	0.95	0.92	0.86	0.89	0.91	0.9175	0.921071
VQC, ZFeatureMap, EfficientSU2, COBYLA, statevector-simulator	1.0	0.95	0.9	0.9	0.92	0.893333	0.88	0.920476
RandomSearch, Lasso, results	1.0	1.0	0.94	0.82	0.895	0.893333	0.89	0.919762
VQC, ZFeatureMap, EfficientSU2, COBYLA, qasm-simulator	1.0	0.95	0.9	0.88	0.92	0.91	0.845	0.915
QKE, PauliFeatureMap, aer-simulator, 1.0	0.9	0.95	0.92	0.89	0.89	0.93	0.91	0.912857
VQC, ZFeatureMap, EfficientSU2, SPSA, qasm-simulator	1.0	0.95	0.9	0.86	0.925	0.91	0.845	0.912857
VQC, ZFeatureMap, EfficientSU2, COBYLA, aer-simulator	1.0	0.95	0.92	0.88	0.9	0.906667	0.8275	0.912024
VQC, ZFeatureMap, EfficientSU2, SPSA, statevector-simulator	1.0	0.95	0.92	0.87	0.89	0.89	0.835	0.907857
VQC, ZFeatureMap, RealAmplitudes, COBYLA, aer-simulator	1.0	0.95	0.9	0.86	0.905	0.85	0.865	0.904286
RandomSearch, LightGBM, results	0.4	1.0	0.98	0.95	0.93	0.933333	0.9475	0.877262
OutOfTheBox, LightGBM, results	0.4	1.0	0.96	0.94	0.925	0.936667	0.9375	0.87131
VQC, PauliFeatureMap, EfficientSU2, SPSA, qasm-simulator	0.9	0.75	0.9	0.84	0.89	0.86	0.8675	0.858214
VQC, ZFeatureMap, EfficientSU2, NFT, statevector-simulator	1.0	0.95	0.86	0.72	0.9	0.776667	0.77	0.85381
QKE, PauliFeatureMap, aer-simulator, 31.622776601683793	1.0	0.85	0.96	0.7	0.875	0.826667	0.735	0.849524
QKE, ZFeatureMap, aer-simulator, 31.622776601683793	1.0	1.0	0.88	0.62	0.835	0.736667	0.7475	0.83131
QKE, PauliFeatureMap, aer-simulator, 1000.0	1.0	0.85	0.96	0.58	0.87	0.826667	0.665	0.821667
VQC, PauliFeatureMap, EfficientSU2, SPSA, aer-simulator	0.8	0.75	0.9	0.73	0.845	0.86	0.8525	0.819643
VQC, PauliFeatureMap, EfficientSU2, NFT, statevector-simulator	0.8	0.65	0.9	0.8	0.84	0.783333	0.8475	0.802976
QKE, ZFeatureMap, qasm-simulator, 177.82794100389228	0.9	1.0	0.88	0.57	0.875	0.73	0.6375	0.798929
VQC, ZZFeatureMap, EfficientSU2, COBYLA, aer-simulator	0.7	0.7	0.9	0.71	0.82	0.826667	0.835	0.784524
VQC, ZZFeatureMap, RealAmplitudes, COBYLA, qasm-simulator	0.8	0.7	0.9	0.62	0.775	0.816667	0.785	0.770952
VQC, ZZFeatureMap, RealAmplitudes, NFT, qasm-simulator	0.7	0.7	0.9	0.86	0.775	0.786667	0.535	0.750952
VQC, PauliFeatureMap, RealAmplitudes, NFT, qasm-simulator	0.6	0.7	0.9	0.49	0.8	0.763333	0.78	0.719048
VQC, ZZFeatureMap, RealAmplitudes, COBYLA, aer-simulator	0.5	0.65	0.84	0.73	0.83	0.83	0.575	0.707857
QKE, PauliFeatureMap, aer-simulator, 0.03162277660168379	0.4	0.35	0.9	0.65	0.86	0.923333	0.8275	0.701548
QKE, PauliFeatureMap, aer-simulator, 0.005623413251903491	0.4	0.35	0.9	0.49	0.75	0.766667	0.8275	0.640595
QKE, PauliFeatureMap, qasm-simulator, 0.005623413251903491	0.4	0.35	0.9	0.49	0.75	0.766667	0.8275	0.640595
QKE, ZFeatureMap, statevector-simulator, 0.005623413251903491	0.4	0.35	0.84	0.49	0.63	0.85	0.83	0.627143
VQC, ZFeatureMap, TwoLocal, SPSA, statevector-simulator	0.7	0.65	0.52	0.51	0.52	0.493333	0.58	0.567619
QKE, PauliFeatureMap, qasm-simulator, 0.001	0.4	0.35	0.9	0.49	0.48	0.753333	0.4975	0.552976
OutOfTheBox, Lasso, results	0.4	0.35	0.5	0.49	0.48	0.506667	0.4975	0.460595
VQC, ZZFeatureMap, TwoLocal, COBYLA, qasm-simulator	0.2	0.35	0.28	0.35	0.225	0.216667	0.3975	0.288452
VQC, PauliFeatureMap, TwoLocal, SPSA, qasm-simulator	0.2	0.35	0.26	0.38	0.185	0.223333	0.4	0.285476
VQC, PauliFeatureMap, TwoLocal, COBYLA, statevector-simulator	0.2	0.35	0.28	0.36	0.19	0.223333	0.39	0.284762
VQC, PauliFeatureMap, TwoLocal, SPSA, statevector-simulator	0.2	0.35	0.28	0.36	0.19	0.223333	0.39	0.284762

**Table 2 entropy-25-00992-t002:** This table presents the runtimes of our experiments conducted on artificially generated classification datasets of varying sizes, e.g., 50 and 100. Given these different dataset sizes, this table is sorted in increasing order of the average runtime over all different sample sizes of each algorithm. The measured runtime includes hyperparameter tuning via randomized search and five-fold cross-validating, training, and testing the model. The parametrization for the QKE is as follows: QKE, feature map, quantum simulator, C-Value for the SVM algorithm. The parametrization for the VQC is as follows: VQC, feature map, Ansatz, optimizer, quantum simulator. For the classical machine learning algorithms, *OutOfTheBox* means that we did not tune the hyperparameters of the employed algorithm and *RandomSearch* refers to hyperparameter optimization via a randomized search.

Algorithm/Parametrization	Size 50	Size 100	Size 250	Size 500	Size 1000	Size 1500	Size 2000	Average
OutOfTheBox, Lasso, results	0.001473	0.001162	0.001158	0.001123	0.001141	0.001153	0.001159	0.001196
OutOfTheBox, Ridge, results	0.002933	0.001553	0.001433	0.001894	0.002628	0.002575	0.002436	0.002207
OutOfTheBox, SVM, results	0.001021	0.000648	0.001039	0.002457	0.005501	0.017243	0.0295	0.008201
OutOfTheBox, XGBoost, results	0.016881	0.017187	0.022922	0.038751	0.05111	0.151807	0.120973	0.059947
OutOfTheBox, LightGBM, results	0.009655	0.024887	0.104107	0.124862	0.1898	0.489043	0.218343	0.165814
RandomSearch, Lasso, results	1.045328	0.113413	0.102258	0.105736	0.104031	0.120507	0.116006	0.243897
RandomSearch, Ridge, results	1.120708	0.122188	0.114706	0.175996	0.226949	0.255845	0.25067	0.323866
RandomSearch, SVM, results	1.06376	0.135593	0.163875	0.159699	0.203163	0.354172	0.442741	0.360429
OutOfTheBox, MLP, results	0.082953	0.091169	0.121317	0.232771	0.451674	0.947373	1.376965	0.472032
OutOfTheBox, CatBoost, results	0.389826	0.411965	0.654325	0.783825	0.867595	1.085298	1.1931	0.769419
RandomSearch, LightGBM, results	1.711872	0.376494	0.58387	0.704715	0.728305	0.897428	1.000039	0.857532
RandomSearch, XGBoost, results	1.572541	0.399174	0.441059	0.577969	0.99776	1.467667	1.352474	0.972663
VQC, ZFeatureMap, TwoLocal, SPSA, statevector-simulator	0.502447	0.82391	1.319602	2.9078	6.75953	11.81601	18.064725	6.027718
VQC, PauliFeatureMap, TwoLocal, COBYLA, statevector-simulator	0.536454	0.886945	1.757877	3.486975	8.137821	14.688881	22.79476	7.469959
VQC, PauliFeatureMap, TwoLocal, SPSA, statevector-simulator	1.981785	0.715829	1.621059	3.488372	8.517624	15.170185	22.300972	7.685118
VQC, PauliFeatureMap, TwoLocal, SPSA, qasm-simulator	0.750719	1.154406	2.53449	5.000262	11.265137	19.493945	29.031463	9.89006
VQC, ZZFeatureMap, TwoLocal, COBYLA, qasm-simulator	0.734865	1.097202	2.514703	4.990832	11.895971	19.283406	29.318269	9.976464
RandomSearch, MLP, results	3.899634	3.298337	5.003618	9.651274	14.729924	20.652249	31.202069	12.633872
QKE, ZFeatureMap, statevector-simulator, 0.1778279410038923	1.343983	0.802286	2.170829	5.965899	18.504546	36.659922	59.889941	17.905344
QKE, ZFeatureMap, statevector-simulator, 0.005623413251903491	0.411296	0.697461	2.154164	6.122564	19.670819	37.297334	62.1901	18.363391
QKE, PauliFeatureMap, statevector-simulator, 1000.0	0.470933	0.956269	2.721257	7.2817	21.356298	40.130716	67.422908	20.048583
QKE, PauliFeatureMap, statevector-simulator, 5.623413251903491	0.501446	0.922237	2.775664	7.454642	21.780637	40.426036	66.758927	20.088513
QKE, ZFeatureMap, statevector-simulator, 5.623413251903491	0.378018	0.757363	2.141677	4.962464	19.901565	41.913003	71.453831	20.215417
QKE, ZZFeatureMap, statevector-simulator, 31.622776601683793	0.214386	0.567282	1.650304	5.302437	20.77629	42.614517	72.871078	20.570899
QKE, ZZFeatureMap, statevector-simulator, 177.82794100389228	0.461093	0.943574	2.780804	7.580857	22.906811	41.955521	68.045553	20.667745
RandomSearch, CatBoost, results	8.627878	10.873142	26.728395	35.20857	36.902272	56.253265	37.994929	30.369779
VQC, ZFeatureMap, RealAmplitudes, COBYLA, aer-simulator	47.438183	63.446748	192.148143	404.233954	1060.291657	1619.397205	2290.222381	811.025467
VQC, ZZFeatureMap, RealAmplitudes, COBYLA, qasm-simulator	43.113636	83.175558	166.040938	421.278374	1064.238564	1702.893006	2719.340939	885.725859
VQC, ZZFeatureMap, RealAmplitudes, COBYLA, aer-simulator	45.909504	83.201411	152.20265	509.1956	1158.902532	1654.065907	2603.942577	886.774312
VQC, ZFeatureMap, EfficientSU2, COBYLA, statevector-simulator	48.546243	81.030425	190.958188	402.121722	1044.855825	1807.676357	2751.241623	903.775769
VQC, ZFeatureMap, EfficientSU2, COBYLA, aer-simulator	57.728111	100.590997	240.174666	507.58709	1253.080578	2139.855218	3196.07247	1070.727019
VQC, ZFeatureMap, EfficientSU2, COBYLA, qasm-simulator	59.058898	100.862056	242.285405	507.171731	1262.650143	2151.503499	3191.745568	1073.611043
VQC, ZZFeatureMap, EfficientSU2, COBYLA, aer-simulator	59.651649	105.629842	254.918442	601.245125	1335.017904	2260.354294	3366.65501	1140.496038
QKE, ZFeatureMap, qasm-simulator, 177.82794100389228	4.589478	13.184805	82.633779	332.71327	1337.102907	3020.689579	5368.201509	1451.30219
QKE, ZZFeatureMap, qasm-simulator, 5.623413251903491	4.352785	15.921249	97.165028	390.472092	1573.103197	3549.629798	6282.670251	1701.902057
QKE, PauliFeatureMap, aer-simulator, 0.03162277660168379	3.549125	15.094144	98.970568	393.496921	1581.662241	3554.962927	6317.355669	1709.298799
QKE, PauliFeatureMap, aer-simulator, 0.005623413251903491	3.373257	15.311538	99.2351	390.52131	1574.108371	3555.3048	6339.026443	1710.982974
QKE, PauliFeatureMap, qasm-simulator, 0.005623413251903491	3.812115	19.479307	101.289711	404.432384	1636.24686	3642.937393	6307.605039	1730.828973
QKE, PauliFeatureMap, aer-simulator, 31.622776601683793	3.848578	17.062982	101.387533	408.69903	1635.863136	3674.976257	6555.811507	1771.092718
VQC, ZFeatureMap, EfficientSU2, NFT, statevector-simulator	98.831974	167.48274	394.378037	836.913451	2197.652135	3719.047116	5621.134708	1862.205737
VQC, PauliFeatureMap, EfficientSU2, NFT, statevector-simulator	103.914165	177.047181	423.423603	1014.963511	2338.078356	3953.861723	5905.433094	1988.10309
VQC, ZZFeatureMap, RealAmplitudes, NFT, qasm-simulator	105.987181	183.918751	427.016702	1036.605473	2366.463152	4052.521035	6042.538015	2030.721473
VQC, PauliFeatureMap, RealAmplitudes, NFT, qasm-simulator	103.625823	180.306618	425.488049	1041.160999	2371.366715	4044.856475	6048.573929	2030.768373
VQC, ZFeatureMap, EfficientSU2, SPSA, statevector-simulator	119.513477	200.101417	474.113288	1008.932874	2601.731917	4505.306268	6781.089745	2241.541284
VQC, ZFeatureMap, EfficientSU2, SPSA, qasm-simulator	145.295744	256.711762	609.791229	1272.675059	3150.527537	5366.116602	8009.649075	2687.25243
VQC, PauliFeatureMap, EfficientSU2, SPSA, aer-simulator	144.280811	259.102175	625.193096	1502.476923	3356.340799	5689.827615	8454.144295	2861.623673
VQC, PauliFeatureMap, EfficientSU2, SPSA, qasm-simulator	152.666649	269.680847	642.400747	1505.762521	3388.662998	5709.505826	8438.957709	2872.519614
QKE, ZFeatureMap, aer-simulator, 31.622776601683793	5.993241	25.852654	166.703792	669.201309	2934.169598	6729.31411	12,037.430687	3224.095056
QKE, PauliFeatureMap, qasm-simulator, 5.623413251903491	8.384715	32.795287	206.595473	890.414904	3753.488868	8537.768589	15,232.745542	4094.599054
QKE, PauliFeatureMap, qasm-simulator, 0.001	7.792093	32.566225	207.832614	896.042249	3778.324351	8610.335147	15,348.810142	4125.957546
QKE, ZFeatureMap, aer-simulator, 0.1778279410038923	10.511296	43.335078	276.810734	1111.545614	4799.032996	10,979.135601	19,768.073574	5284.063556
QKE, ZFeatureMap, qasm-simulator, 5.623413251903491	11.573929	43.186982	277.291314	1113.664313	4842.587094	10,978.908476	19,798.821156	5295.147609
QKE, PauliFeatureMap, aer-simulator, 1000.0	12.596938	51.788837	332.281104	1434.208601	5986.631006	13,592.866065	24,280.544075	6527.273804
QKE, PauliFeatureMap, aer-simulator, 1.0	12.261604	51.508959	332.561822	1423.111135	5984.902587	13,603.956887	24,362.83202	6538.733573

**Table 3 entropy-25-00992-t003:** This table presents the scores/accuracies of our experiments conducted on classification datasets generated via SU(2) generators of varying sizes, e.g., 50 and 100. Given these different dataset sizes, this table is sorted in decreasing order of the average accuracy over all different sample sizes of each algorithm. The parametrization for the QKE is as follows: QKE, feature map, quantum simulator, C-Value for the SVM algorithm. The parametrization for the VQC is as follows: VQC, feature map, Ansatz, optimizer, quantum simulator. For the classical machine learning algorithms, *OutOfTheBox* means that we did not tune the hyperparameters of the employed algorithm and *RandomSearch* refers to hyperparameter optimization via a randomized search.

Algorithm/Parametrization	Size 50	Size 100	Size 250	Size 500	Size 1000	Size 1500	Size 2000	Average
RandomSearch, CatBoost, results	0.9	0.55	0.78	0.78	0.88	0.906667	0.915	0.815952
OutOfTheBox, CatBoost, results	0.6	0.7	0.76	0.85	0.895	0.906667	0.9375	0.807024
RandomSearch, XGBoost, results	0.6	0.65	0.74	0.84	0.87	0.86	0.9425	0.786071
OutOfTheBox, XGBoost, results	0.4	0.75	0.76	0.86	0.89	0.926667	0.9	0.78381
QKE, ZFeatureMap, statevector-simulator, 1000.0	0.7	0.8	0.74	0.79	0.8	0.806667	0.8475	0.783452
RandomSearch, LightGBM, results	0.3	0.6	0.74	0.87	0.89	0.903333	0.9175	0.745833
OutOfTheBox, LightGBM, results	0.3	0.6	0.74	0.9	0.85	0.923333	0.885	0.742619
QKE, ZZFeatureMap, statevector-simulator, 177.82794100389228	0.8	0.55	0.54	0.71	0.795	0.826667	0.85	0.724524
QKE, PauliFeatureMap, aer-simulator, 5.623413251903491	0.7	0.65	0.44	0.73	0.725	0.683333	0.72	0.664048
QKE, ZZFeatureMap, qasm-simulator, 31.622776601683793	0.7	0.7	0.7	0.59	0.615	0.62	0.665	0.655714
QKE, ZZFeatureMap, statevector-simulator, 0.1778279410038923	0.8	0.5	0.62	0.55	0.645	0.693333	0.7375	0.649405
QKE, ZZFeatureMap, aer-simulator, 0.1778279410038923	0.4	0.6	0.7	0.67	0.625	0.713333	0.7075	0.630833
QKE, PauliFeatureMap, aer-simulator, 0.1778279410038923	0.5	0.65	0.64	0.63	0.615	0.68	0.7	0.630714
QKE, PauliFeatureMap, statevector-simulator, 0.1778279410038923	0.4	0.65	0.6	0.55	0.75	0.666667	0.73	0.620952
OutOfTheBox, MLP, results	0.8	0.55	0.52	0.63	0.555	0.56	0.62	0.605
VQC, ZZFeatureMap, EfficientSU2, SPSA, aer-simulator	0.7	0.5	0.6	0.6	0.605	0.576667	0.645	0.60381
VQC, ZZFeatureMap, EfficientSU2, COBYLA, qasm-simulator	0.6	0.6	0.62	0.54	0.6	0.606667	0.66	0.60381
OutOfTheBox, SVM, results	0.4	0.65	0.44	0.59	0.68	0.676667	0.695	0.590238
RandomSearch, SVM, results	0.2	0.7	0.8	0.65	0.78	0.503333	0.4725	0.586548
VQC, ZZFeatureMap, EfficientSU2, NFT, statevector-simulator	0.5	0.6	0.54	0.56	0.7	0.58	0.61	0.584286
VQC, ZZFeatureMap, RealAmplitudes, COBYLA, statevector-simulator	0.6	0.65	0.6	0.53	0.575	0.543333	0.5775	0.582262
VQC, PauliFeatureMap, EfficientSU2, COBYLA, aer-simulator	0.4	0.75	0.52	0.55	0.56	0.606667	0.6225	0.572738
VQC, PauliFeatureMap, RealAmplitudes, NFT, statevector-simulator	0.3	0.65	0.52	0.69	0.63	0.586667	0.62	0.570952
OutOfTheBox, Ridge, results	0.7	0.55	0.62	0.48	0.575	0.533333	0.525	0.569048
VQC, ZFeatureMap, RealAmplitudes, COBYLA, qasm-simulator	0.7	0.5	0.52	0.57	0.59	0.573333	0.5275	0.56869
VQC, ZZFeatureMap, EfficientSU2, NFT, aer-simulator	0.4	0.7	0.56	0.58	0.575	0.543333	0.6175	0.567976
QKE, ZZFeatureMap, aer-simulator, 31.622776601683793	0.4	0.55	0.6	0.62	0.625	0.596667	0.555	0.56381
QKE, PauliFeatureMap, aer-simulator, 31.622776601683793	0.6	0.45	0.6	0.56	0.57	0.62	0.5425	0.563214
VQC, ZZFeatureMap, RealAmplitudes, COBYLA, aer-simulator	0.6	0.65	0.48	0.55	0.535	0.573333	0.545	0.561905
QKE, ZFeatureMap, qasm-simulator, 177.82794100389228	0.6	0.7	0.5	0.52	0.5	0.556667	0.53	0.558095
VQC, ZFeatureMap, EfficientSU2, COBYLA, aer-simulator	0.7	0.6	0.5	0.48	0.525	0.48	0.615	0.557143
VQC, ZZFeatureMap, RealAmplitudes, NFT, qasm-simulator	0.6	0.4	0.58	0.47	0.66	0.573333	0.61	0.55619
VQC, ZFeatureMap, EfficientSU2, COBYLA, qasm-simulator	0.7	0.5	0.56	0.55	0.51	0.566667	0.505	0.555952
QKE, PauliFeatureMap, qasm-simulator, 0.1778279410038923	0.5	0.35	0.36	0.63	0.655	0.693333	0.7025	0.555833
QKE, ZZFeatureMap, qasm-simulator, 0.001	0.7	0.45	0.66	0.57	0.54	0.466667	0.4725	0.55131
VQC, ZFeatureMap, RealAmplitudes, SPSA, aer-simulator	0.7	0.5	0.5	0.59	0.495	0.516667	0.555	0.550952
VQC, ZFeatureMap, EfficientSU2, SPSA, statevector-simulator	0.3	0.8	0.44	0.59	0.55	0.563333	0.595	0.548333
VQC, PauliFeatureMap, RealAmplitudes, NFT, qasm-simulator	0.4	0.55	0.6	0.54	0.59	0.553333	0.58	0.544762
VQC, ZFeatureMap, RealAmplitudes, COBYLA, aer-simulator	0.8	0.45	0.48	0.48	0.52	0.513333	0.545	0.54119
VQC, ZFeatureMap, EfficientSU2, COBYLA, statevector-simulator	0.5	0.55	0.56	0.56	0.54	0.516667	0.56	0.540952
QKE, ZFeatureMap, aer-simulator, 1000.0	0.6	0.55	0.6	0.52	0.505	0.486667	0.52	0.540238
VQC, PauliFeatureMap, EfficientSU2, SPSA, qasm-simulator	0.4	0.45	0.56	0.64	0.575	0.533333	0.6075	0.537976
VQC, ZFeatureMap, RealAmplitudes, NFT, aer-simulator	0.5	0.4	0.62	0.52	0.58	0.533333	0.5875	0.534405
VQC, PauliFeatureMap, EfficientSU2, SPSA, aer-simulator	0.4	0.45	0.54	0.54	0.64	0.56	0.6	0.532857
QKE, ZFeatureMap, statevector-simulator, 0.1778279410038923	0.5	0.45	0.54	0.54	0.585	0.556667	0.555	0.532381
QKE, ZFeatureMap, aer-simulator, 1.0	0.4	0.5	0.4	0.63	0.56	0.603333	0.575	0.524048
QKE, ZFeatureMap, qasm-simulator, 1000.0	0.6	0.5	0.44	0.59	0.525	0.443333	0.5	0.514048
QKE, PauliFeatureMap, statevector-simulator, 0.03162277660168379	0.5	0.45	0.58	0.56	0.46	0.476667	0.5675	0.513452
RandomSearch, MLP, results	0.3	0.25	0.5	0.6	0.635	0.64	0.6675	0.513214
QKE, ZFeatureMap, aer-simulator, 177.82794100389228	0.4	0.65	0.54	0.46	0.495	0.48	0.565	0.512857
QKE, ZFeatureMap, qasm-simulator, 0.005623413251903491	0.5	0.65	0.48	0.47	0.515	0.48	0.485	0.511429
RandomSearch, Ridge, results	0.4	0.55	0.52	0.46	0.56	0.52	0.5675	0.511071
OutOfTheBox, Lasso, results	0.5	0.7	0.4	0.51	0.495	0.46	0.495	0.508571
RandomSearch, Lasso, results	0.6	0.45	0.44	0.45	0.515	0.533333	0.54	0.504048

**Table 4 entropy-25-00992-t004:** This table presents the scores/accuracies of our experiments conducted on classification datasets generated via SU(2) generators of varying sizes, e.g., 50 and 100. Given these different dataset sizes, this table is sorted in increasing order of the average runtime over all different sample sizes of each algorithm. The measured runtime includes hyperparameter tuning via randomized search and five-fold cross-validating, training, and testing the model. The parametrization for the QKE is as follows: QKE, feature map, quantum simulator, C-Value for the SVM algorithm. The parametrization for the VQC is as follows: VQC, feature map, Ansatz, optimizer, quantum simulator. For the classical machine learning algorithms, *OutOfTheBox* means that we did not tune the hyperparameters of the employed algorithm and *RandomSearch* refers to hyperparameter optimization via a randomized search.

Algorithm/Parametrization	Size 50	Size 100	Size 250	Size 500	Size 1000	Size 1500	Size 2000	Average
OutOfTheBox, Lasso, results	0.004103	0.000646	0.000661	0.001249	0.000804	0.000694	0.000691	0.001264
OutOfTheBox, Ridge, results	0.003733	0.002898	0.002786	0.029688	0.001899	0.001929	0.00178	0.006388
OutOfTheBox, SVM, results	0.00111	0.000919	0.002298	0.006078	0.012139	0.025935	0.04667	0.013593
RandomSearch, Lasso, results	1.055654	0.123047	0.103457	0.104252	0.117591	0.124301	0.115839	0.249163
RandomSearch, Ridge, results	1.084348	0.122741	0.138248	0.145562	0.156783	0.129915	0.14262	0.274317
OutOfTheBox, XGBoost, results	0.026616	0.040969	0.207265	1.348421	0.190274	0.123878	0.205496	0.306131
RandomSearch, SVM, results	1.133957	0.167929	0.131167	0.147877	0.233314	0.444834	0.438209	0.385327
OutOfTheBox, CatBoost, results	1.072432	0.439886	0.68483	0.813699	0.936663	1.13145	1.252242	0.904457
RandomSearch, XGBoost, results	1.555324	0.398761	0.517864	0.817118	1.79372	2.056436	2.289563	1.346969
OutOfTheBox, LightGBM, results	0.134499	0.935128	0.319397	1.125076	5.646488	4.908206	4.312101	2.482985
OutOfTheBox, MLP, results	0.447343	0.385258	0.303056	1.874662	2.165486	5.582801	6.986128	2.534962
RandomSearch, LightGBM, results	3.76752	0.726469	0.892496	1.349411	5.942432	5.569574	3.741842	3.141392
QKE, ZFeatureMap, statevector-simulator, 1000.0	0.588803	1.105112	3.115228	9.692818	24.488245	46.863381	75.605305	23.065556
QKE, ZZFeatureMap, statevector-simulator, 177.82794100389228	1.023377	2.014738	6.133744	14.08855	36.209869	63.510698	97.575217	31.508028
QKE, PauliFeatureMap, statevector-simulator, 0.1778279410038923	1.110054	2.154783	6.591036	14.487802	36.089562	62.912848	97.64273	31.569831
RandomSearch, MLP, results	16.90452	15.129156	5.671446	42.576796	38.564641	70.807035	64.602048	36.322235
QKE, ZFeatureMap, statevector-simulator, 0.1778279410038923	1.235019	1.984607	5.547594	15.333808	41.977686	79.052726	127.710628	38.977438
QKE, PauliFeatureMap, statevector-simulator, 0.03162277660168379	1.453579	2.804056	9.085313	19.440721	47.633347	81.740004	128.186636	41.477665
QKE, ZZFeatureMap, statevector-simulator, 0.1778279410038923	2.194236	4.856549	10.043207	20.391739	57.658834	97.515617	151.070676	49.104408
RandomSearch, CatBoost, results	18.350174	35.654742	40.725868	70.65956	68.788446	68.958949	43.685619	49.546194
VQC, ZFeatureMap, RealAmplitudes, COBYLA, qasm-simulator	55.500573	100.423827	241.59017	577.400258	1286.962867	2164.282246	3315.185871	1105.906545
VQC, ZFeatureMap, RealAmplitudes, COBYLA, aer-simulator	61.174412	116.296597	274.944359	672.215972	1509.987298	2624.710516	4058.819283	1331.164062
VQC, ZZFeatureMap, RealAmplitudes, COBYLA, statevector-simulator	68.600875	123.207802	380.446356	770.716185	1635.89919	2621.242822	3805.021206	1343.590634
VQC, PauliFeatureMap, EfficientSU2, COBYLA, aer-simulator	89.832315	163.70887	480.279572	975.291255	2084.174564	3407.844934	5050.433405	1750.223559
VQC, ZZFeatureMap, EfficientSU2, COBYLA, qasm-simulator	88.21128	163.988243	480.35886	973.614566	2068.950356	3425.564759	5057.849761	1751.219689
VQC, ZFeatureMap, EfficientSU2, COBYLA, qasm-simulator	85.847133	156.496491	381.392026	888.461174	2300.527629	3878.06995	6136.175247	1975.281379
VQC, ZFeatureMap, EfficientSU2, COBYLA, statevector-simulator	103.51928	191.066923	456.017277	1079.39181	2305.444005	3940.212229	5958.277453	2004.846997
VQC, ZFeatureMap, RealAmplitudes, NFT, aer-simulator	111.03018	203.235006	491.978903	1181.668828	2620.544781	4428.84701	6770.538213	2258.263274
VQC, ZFeatureMap, EfficientSU2, COBYLA, aer-simulator	113.765615	205.589837	516.289881	1202.288662	2663.121947	4533.378097	6730.84663	2280.754381
VQC, ZZFeatureMap, RealAmplitudes, COBYLA, aer-simulator	111.686165	209.034186	638.294179	1296.48554	2869.708946	4781.981067	7074.925479	2426.016509
VQC, PauliFeatureMap, RealAmplitudes, NFT, qasm-simulator	163.24924	303.991817	936.898012	1915.698994	4145.466983	6900.43208	10,273.374065	3519.873027
VQC, ZFeatureMap, EfficientSU2, SPSA, statevector-simulator	190.048137	341.454534	823.875859	1930.637818	4188.237967	6999.258807	10,586.6448	3580.02256
VQC, PauliFeatureMap, RealAmplitudes, NFT, statevector-simulator	190.485358	349.906865	1108.180672	2248.345232	4814.143806	7883.610885	11,821.059517	4059.390334
VQC, ZFeatureMap, RealAmplitudes, SPSA, aer-simulator	195.45132	357.101921	856.679886	2110.857992	4766.449826	8178.593323	12,549.799371	4144.99052
VQC, ZZFeatureMap, RealAmplitudes, NFT, qasm-simulator	224.602174	405.497928	1270.741552	2674.141109	5725.217252	9676.191547	14,306.04881	4897.491482
VQC, ZZFeatureMap, EfficientSU2, NFT, statevector-simulator	243.54281	457.172266	1372.528644	2784.422174	5896.297166	9666.126191	14,179.331314	4942.774366
QKE, ZFeatureMap, aer-simulator, 1.0	9.847104	40.399745	258.379268	1171.612393	4847.638381	10,994.736629	19,569.132042	5270.249366
VQC, ZZFeatureMap, EfficientSU2, NFT, aer-simulator	259.240503	473.126474	1408.601057	2903.563737	6301.786963	10,332.959201	15,345.314436	5289.227482
VQC, PauliFeatureMap, EfficientSU2, SPSA, aer-simulator	316.140029	574.301472	1727.700192	3530.722918	6706.076042	9861.491595	14,726.388476	5348.974389
QKE, ZFeatureMap, aer-simulator, 1000.0	10.898171	46.773357	297.390107	1339.784466	5587.107904	11,607.623344	19,440.97092	5475.79261
VQC, ZZFeatureMap, EfficientSU2, SPSA, aer-simulator	243.897139	463.586231	1368.88324	2789.133991	6570.676821	11,456.705765	17,726.796588	5802.811396
VQC, PauliFeatureMap, EfficientSU2, SPSA, qasm-simulator	348.216144	639.131772	1898.469802	3945.807249	8437.848893	13,815.446673	20,511.034176	7085.136387
QKE, ZFeatureMap, qasm-simulator, 1000.0	11.579451	47.675482	344.543775	1568.903939	6191.830912	14,908.467701	27,002.766078	7153.681048
QKE, ZFeatureMap, aer-simulator, 177.82794100389228	14.163619	56.856619	359.793343	1620.482626	6647.990849	15,084.137764	26,961.326713	7249.250219
QKE, ZFeatureMap, qasm-simulator, 177.82794100389228	16.35717	77.129608	482.478877	2237.68152	9219.954344	18,899.046552	26,623.312487	8222.28008
QKE, ZFeatureMap, qasm-simulator, 0.005623413251903491	16.184459	68.030962	439.889123	2003.14586	8339.157072	18,939.866418	33,875.189822	9097.351959
QKE, PauliFeatureMap, aer-simulator, 31.622776601683793	16.822446	70.391611	549.285996	2267.794391	9148.306499	20,490.131389	36,687.638808	9890.05302
QKE, ZZFeatureMap, aer-simulator, 31.622776601683793	17.382234	70.921393	552.720236	2290.305118	9223.01824	20,681.450668	36,991.554065	9975.335993
QKE, ZZFeatureMap, aer-simulator, 0.1778279410038923	19.618006	80.653612	632.012298	2628.407038	9714.431489	20,666.725844	36,766.378776	10,072.603866
QKE, PauliFeatureMap, aer-simulator, 5.623413251903491	20.03461	81.805468	657.437384	2646.600018	10,751.043722	24,303.410594	42,050.186601	11,501.502628
QKE, ZZFeatureMap, qasm-simulator, 0.001	22.474871	94.5939	748.53639	3061.492908	11,095.037557	24,179.108494	42,833.544061	11,719.255454
QKE, PauliFeatureMap, aer-simulator, 0.1778279410038923	15.70449	64.293166	539.372432	2052.767245	10,360.381735	28,219.134103	53,610.138579	13,551.684536
QKE, PauliFeatureMap, qasm-simulator, 0.1778279410038923	28.777769	121.675706	961.248534	3951.052992	16,159.343561	35,692.431334	48,691.262451	15,086.541764
QKE, ZZFeatureMap, qasm-simulator, 31.622776601683793	28.201021	110.795119	877.141222	3647.413017	16,257.207805	38,819.796065	69,300.661749	18,434.459428

**Table 5 entropy-25-00992-t005:** These tables present the scores/accuracies of our experiments conducted on publicly available classification datasets. The upper table displays the best five-fold cross-validation scores, obtained using randomized search cross-validation from Scikit-learn, which were employed to identify the optimal model. The lower table shows the scores of the best model evaluated on an unseen test subset of the original data. We include results for the six datasets described in Section 3.5, the quantum classifiers detailed in Section 3.3, and the classical machine learning classifiers discussed in Section 3.2.

Classifier\Dataset	Iris	Wine	ILPD	BC-Coimbra	TAE	Breast-Tissue
**VQC**	0.817	0.817	0.706	0.599	0.417	0.339
**QKE**	0.908	0.853	0.706	0.620	0.483	0.382
**Ridge**	0.914	0.875	0.080	0.053	0.053	<0.001
**Lasso**	0.914	0.870	0.085	0.004	0.004	<0.001
**MLP**	**0.975**	0.937	**0.712**	0.687	0.425	0.406
**SVM**	0.958	0.759	0.706	0.630	0.450	0.382
**XGBoost**	0.958	**0.986**	0.695	0.656	**0.533**	**0.441**
**LightGBM**	0.967	**0.986**	0.699	0.666	0.475	0.393
**CatBoost**	0.950	0.979	0.702	**0.688**	0.525	0.440

**Classifier\Dataset**	**Iris**	**Wine**	**ILPD**	**BC-Coimbra**	**TAE**	**Breast-Tissue**
**VQC**	0.767	0.639	0.744	0.541	0.388	0.334
**QKE**	**1.0**	0.833	0.744	0.792	0.613	0.409
**Ridge**	0.947	0.878	0.115	0.234	<0.001	<0.001
**Lasso**	0.945	0.882	0.115	0.296	<0.001	<0.001
**MLP**	**1.0**	**1.0**	**0.769**	0.875	0.387	0.455
**SVM**	**1.0**	0.972	0.743	0.875	0.355	0.455
**XGBoost**	**1.0**	**1.0**	0.735	**0.917**	0.533	0.441
**LightGBM**	**1.0**	**1.0**	0.752	**0.917**	0.419	0.455
**CatBoost**	**1.0**	**1.0**	0.744	**0.917**	**0.645**	**0.545**

**Table 6 entropy-25-00992-t006:** This table presents the combined runtimes of our experiments conducted on well-known and publicly available classification datasets. The runtimes include both the five-fold randomized search cross-validation process from Scikit-learn, which was employed to identify the optimal model, and the evaluation of the best model on an unseen test subset of the original data. We include results for the six datasets described in Section 3.5, the quantum classifiers detailed in Section 3.3, and the classical machine learning classifiers discussed in Section 3.2.

Classifier\ Dataset	Iris	Wine	ILPD	BC-Coimbra	TAE	Breast-Tissue
**VQC**	3:32:16.547605	1 day, 13:56:59.455185	2 days, 23:03:26.398856	9:55:17.907443	2:46:25.921553	9:01:58.623806
**QKE**	2:03:57.921154	21:41:38.738255	7 days, 6:30:41.179676	5:02:26.430001	1:28:54.069725	3:37:05.655104
**Ridge**	0:00:00.175009	0:00:00.496771	0:00:00.399229	0:00:00.240857	0:00:00.209600	0:00:00.296966
**Lasso**	0:00:00.173051	0:00:00.181444	0:00:00.237455	0:00:00.192257	0:00:00.229508	0:00:00.225531
**MLP**	0:00:16.876288	0:00:10.477420	0:00:26.748907	0:00:10.951229	0:00:08.475263	0:00:13.729790
**SVM**	0:00:00.143353	0:00:00.165431	0:00:00.484485	0:00:00.180694	0:00:00.228508	0:00:00.226784
**XGBoost**	0:00:03.809085	0:00:04.030425	0:00:04.752627	0:00:02.744122	0:00:05.820371	0:00:06.864497
**LightGBM**	0:00:02.971164	0:00:03.180770	0:00:03.062553	0:00:01.462174	0:00:03.056615	0:00:04.540870
**CatBoost**	0:00:06.465975	0:00:18.511612	0:00:11.352944	0:00:07.460460	0:00:06.964821	0:00:26.639070

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
