# Peer review of "On the Applicability of Quantum Machine Learning"

_entropy, 2023, doi:10.3390/e25070992_

Round 1

Reviewer 1 Report

The manuscript entitled «On the Applicability of Quantum Machine Learning» by Sebastian Raubitzek and Kevin Mallinger investigates the applicability of quantum machine learning for classification tasks using two quantum classifiers from the Qiskit Python environment: the Variational Quantum Classifier (VQC) and the Quantum Kernel Estimator (QKE). They evaluated the performance of these classifiers on seven widely known classical machine learning (such as Lasso, Ridge, MLP, SVM, XGBoost, LightGBM, and CatBoost). They used six benchmark datasets and two artificially generated classification datasets. One of the artificial datasets is based on concepts from quantum mechanics using the exponential map of a Lie algebra.

Suggestions:

Question

Line 64: “To what extent can randomized search make the performance of quantum algorithms comparable to classical approaches?” Which randomized search? The wording of the question should be adjusted to align with the obtained results.

 Related work

Additional studies pertaining to the evaluation or application of the Variational Quantum Classifier (VQC) and the Quantum Kernel Estimator (QKE) could be incorporated.

Terms

Sometimes the term "Variational Quantum Classifier (VQC)" is used and other times "Variational Quantum Circuit(VQC)" e.g. Section 2.3.1 . Could you please clarify the relationship between these terms?

Figures

In Figure 2 the word “optimizer” is presented.

Tables

Tables 1, 2, 3, and 4 are generally not clear enough.

In Tables 1 and 2, the classic algorithms are presented as "CatBoost, OutOfTheBox" and "CatBoost, RandomizedSearchCV," respectively. However, in Tables 3 and 4, they are presented as "OutOfTheBox, CatBoost" and "RandomSearch, CatBoost.". Please, ensure consistency in the naming convention across tables unless there is a specific reason for the variation.

In the caption of Tables 3 and 4, it may be appropriate to replace the phrase "artificially generated classification data sets" with "artificially denatured SU2 data sets."

Tables 2 και 4. In the provided code from the GitHub repository https://github.com/Raubkatz/Quantum_Machine_Learning the time presented in Tables 2 and 4 specifically refers to the training time of the classical models (e.g. ext_act\Lasso_artificial_data_set), while in quantum algorithms time refers to training time and testing time (prediction) e.g. (ext_act\VQC_artificial_data_set_random_picks).

a) Would it be appropriate to include testing time for classical models as well, to ensure that all metrics measure the same aspect?

b) Throughout the paper, it is important to explicitly state what the reported runtime refers to.

Tables 2 and 4: In certain cases, it has been observed that the runtime on dataset 50 exceeds the runtime on dataset 2000 (e.g. Table 2 Ridge, RandomizedSearchCV ). Could you please explain the potential reasons?

Running Environment

Line 535 Section 4.3 «Further, everything was calculated on our local machine using an Intel(R) Core(TM) i7-4770 CPU 3.40GHz and 16GB RAM.»  In sections 4.1 and 4.2 you did not mention on which machine the experiments were run. Please state all parameters of the running environment (machine/pc, Operating System, Qiskit version, packages version e.t.c )

Abbreviations

Throughout the text, complete phrases appear many times, for example, classical machine learning (ML), quantum machine learning (QML), etc. It is suggested to employ abbreviations for enhanced readability and conciseness.

Equations

Line 397: Equations 6-9: Could you explain the operators?

Conclusion

Line 576: “Additionally, their longer runtimes and less consistent performance across the benchmark data sets may...” To ensure clarity, it is recommended to explicitly state that the reported runtimes encompass the 5-fold Randomized Search cross-validation time, the training time, and the testing time.

Citations

Particular attention should be devoted to the bibliographic references to ensure their accuracy and relevance.

Line 37, “Variational Quantum Classifier (VQC) [5]” Is [5] correct? In [5] the term "Variational Quantum Classifier" is not mentioned.

Line 184 “Various techniques, such as cross-validation and train-test splits, are often used to obtain reliable performance estimates of the models [26].”  Is [26] correct?

Line 310 “This data set consists of 10 quantitative predictors and a binary dependent variable, indicating the presence or absence of breast cancer [32? ]. ” Is [32] correct?

Line 225 “VQC is a hybrid quantum-classical algorithm that can be viewed as a quantum analog of classical neural networks, specifically the Multilayer Perceptron (MLP) [4].” Is [4] correct?

Line 346: Quantum Machin Learning, Line 152: CV, explain the abbreviation, Line 355: }, Line 394: these, Line 310:[32? ]. Line 232: , Line 400: 1.

Reviewer 2 Report

In general, this is an interesting and a rather well organised article. However, there are few points which should be sorted out before the manuscript could be recommended for publication in the Journal. 

The first problem is that the manuscript looks like a review paper. Clearly, completely different requirements should be raised for a review manuscript. The presentation style is focused on the description of existing quantum technologies and algorithms. It is impossible to understand the contribution of the authors. Therefore, the authors are recommended to split the presentation is separate parts. The authors should describe the existing state of the art, and then should clearly identify their personal contribution beyond the state of the art.

The second problem is related to the scope of this manuscript. What is the role of entropy in this study? In other words, why this manuscript should deserve a publication in this Journal? Are there any connections to the theory, applications, and methodologies directly related to entropy in this study?

Finally, the results of computations with different datasets are interesting but not convincing. It is well known that quantum algorithms do surpass their non-quantum counter-parties. But the authors should pay more attention to the optimal structure of the particular architecture of the discussed algorithms. For instance, the authors pay zero attention to automatic error correction techniques which are vitally necessary in hardware implementations of quantum processes.

A serious revision is recommended before this manuscript could be reconsidered for possible publication in this Journal.   

Round 2

Reviewer 1 Report

The manuscript has been sufficiently improved to warrant publication in Entropy.

Minor editing of English language required.

Reviewer 2 Report

The authors did perform a good revision and did manage to answer all remarks and comments from the reviewers. The manuscript can be recommended for publication in the present form.